# A pig BodyMap transcriptome reveals diverse tissue physiologies and evolutionary dynamics of transcription

Long Jin [1,13], Qianzi Tang [1,13✉], Silu Hu[1,13], Zhongxu Chen[2,13], Xuming Zhou[3,13], Bo Zeng[1], Yuhao Wang[1], Mengnan He[1], Yan Li[1], Lixuan Gui[2], Linyuan Shen[1], Keren Long[1], Jideng Ma[1], Xun Wang[1], Zhengli Chen[4], Yanzhi Jiang[5], Guoqing Tang[1], Li Zhu[1], Fei Liu[6], Bo Zhang[7], Zhiqing Huang [8], Guisen Li [9], Diyan Li [1], Vadim N. Gladyshev [10], Jingdong Yin[11], Yiren Gu[12], Xuewei Li[1] & Mingzhou Li [1✉]

A comprehensive transcriptomic survey of pigs can provide a mechanistic understanding of tissue specialization processes underlying economically valuable traits and accelerate their use as a biomedical model. Here we characterize four transcript types (lncRNAs, TUCPs, miRNAs, and circRNAs) and protein-coding genes in 31 adult pig tissues and two cell lines. We uncover the transcriptomic variability among 47 skeletal muscles, and six adipose depots linked to their different origins, metabolism, cell composition, physical activity, and mitochondrial pathways. We perform comparative analysis of the transcriptomes of seven tissues from pigs and nine other vertebrates to reveal that evolutionary divergence in transcription potentially contributes to lineage-specific biology. Long-range promoter–enhancer interaction analysis in subcutaneous adipose tissues across species suggests evolutionarily stable transcription patterns likely attributable to redundant enhancers buffering gene expression patterns against perturbations, thereby conferring robustness during speciation. This study can facilitate adoption of the pig as a biomedical model for human biology and disease and uncovers the molecular bases of valuable traits.

[1] Institute of Animal Genetics and Breeding, College of Animal Science and Technology, Sichuan Agricultural University, Chengdu, Sichuan, China. [2] Department of Life Science, Tcuni Inc., Chengdu, Sichuan, China. [3] Institute of Zoology, Chinese Academy of Sciences, Beijing, China. [4] Key Laboratory of Animal Disease and Human Health of Sichuan Province, College of Veterinary Medicine, Sichuan Agricultural University, Chengdu, Sichuan, China. [5] College of Life Science, Sichuan Agricultural University, Ya'an, Sichuan, China. [6] Information and Educational Technology Center, Sichuan Agricultural University, Chengdu, Sichuan, China. [7] Ya'an Digital Economy Operation Company, Ya'an, Sichuan, China. [8] Institute of Animal Nutrition, Sichuan Agricultural University, Chengdu, Sichuan, China. [9] Renal Department and Nephrology Institute, Sichuan Provincial People's Hospital, Chengdu, Sichuan, China. [10] Division of Genetics, Department of Medicine, Brigham and Women's Hospital, Boston, MA, USA. [11] State Key Laboratory of Animal Nutrition, College of Animal Science and Technology, China Agricultural University, Beijing, China. [12] Animal Breeding and Genetics Key Laboratory of Sichuan Province, Sichuan Animal Science Academy, Chengdu, Sichuan, China. [13] These authors contributed equally: Long Jin, Qianzi Tang, Silu Hu, Zhongxu Chen, Xuming Zhou. ✉email: tangqianzi@sicau.edu.cn; mingzhou.li@sicau.edu.cn

**S**us scrofa (i.e., pig or swine) is of substantial agricultural value worldwide and is increasingly used as a model for human diseases[1] and as a source of cells and tissues for xenotransplantation[2]. Systematic analysis of the transcriptome is of central importance to the pig research community. With the FANTOM 5 Consortium[3], ENCODE project[4,5], and Genotype-Tissue Expression (GTEx) Consortium[6,7] in the human and mouse; BodyMap database[8] in rats; and the Functional Annotation of Animal Genomes (FAANG) project[9,10] in sheep, comprehensive characterization of the transcriptome has greatly contributed to our understanding of the regulatory mechanisms and genome complexity of mammals. A recently published, high-quality, continuous reference pig assembly (Sscrofa11.1)[11] has provided a framework to obtain a more complete and accurate view of transcribed sequences, comparable to the richly annotated human and biomedically relevant rodent model reference assemblies (Supplementary Fig. 1).

Apart from protein-coding genes (PCGs), which are the main drivers of phenotypes, mammalian genomes encode several other transcript types (including transcripts of unknown coding potential [TUCPs][12], long noncoding RNAs [lncRNAs][12,13], microRNAs [miRNAs][14], and circular RNAs [circRNAs][15,16]), which play essential regulatory roles via diverse mechanisms and underlie the complexity and variation of the transcriptome, as it relates to tissue physiology[17,18]. Systematic analysis of the different components that comprise the pig transcriptome has long been overdue, and elucidation of these different transcripts and their expression patterns may allow the construction of an atlas of regulatory functions for the many different transcripts and interacting gene networks.

To characterize transcriptomic variability with respect to known tissue-specific physiological activities and identify key transcripts underlying economically important phenotypes in pigs (notably, the production of pork for improved nutritional content), we sequenced 194 paired-end rRNA-depleted RNA-seq libraries (~14.39 gigabases [Gb] of high-quality sequences per library; ~2.79 terabases [Tb] total) and 187 single-end small RNA-seq libraries (~11.91 million [M] reads per library; ~2.23 billion reads total) from 70 tissues (1–3 libraries for each of 17 solid organs, as well as 47 skeletal muscle tissues [SMTs] and six adipose tissues [ATs] from different body sites), and two immortalized cell lines (kidney epithelial cells [PK15] and iliac endothelial cells [PIECs]) (Supplementary Data 1–2). We also sequenced 142 rRNA-depleted RNA-seq libraries (~12.25 Gb of high-quality sequences per library; ~1.74 Tb in total) of seven homologous tissues from eight representative mammalian models and a bird (chicken) (Supplementary Data 3), with the goal of examining the evolutionary dynamics of transcription among animal models in a comparative transcriptomic framework.

## Results

**An expanded landscape of the pig transcriptome**. We performed an enhanced annotation of four distinct transcript types (Table 1; Supplementary Figs. 2–8), including 2440 TUCPs (first identified in this study), 19,072 lncRNAs (among which, 12,180, or 63.86%, were not yet been annotated in the reference genome) (Supplementary Fig. 6), 48,232 circRNAs (first identified in this study), and 1245 miRNAs (783, or 62.89%, apparently novel and absent in the miRbase database[19] for pigs) based on 85 RNA-seq libraries and 78 small RNA-seq libraries from 31 tissues and two cell lines (Fig. 1a) (representing the core atlas dataset for a de novo assembly of transcribed sequences, see Methods for details).

Using 21,303 PCGs in the reference pig assembly[11] (in our study, 20,505, or 96.25%, had evidence of transcription [transcripts per million (TPM) ≥0.1 in at least one sample]), we

investigated the initial characteristics (i.e., sequence or exonic structural features) of biogenesis and functionally distinct transcripts, which were highly similar to those in other mammals. Compared with the evolutionarily conserved PCGs (100-vertebrate phastCons = 0.67), the relatively rapidly evolving lncRNAs (phastCons = 0.07) were shorter (1,694 bp vs. 3,259 bp for PCGs) and had fewer exons (~2.63 exons per transcript vs. ~10.21 for PCGs), although their average exon length was not shorter (~339 bp vs. ~129 bp for PCGs) (Table 1, Fig. 1e, and Supplementary Fig. 7a–c). This observation possibly reflects an evolutionary scenario wherein de novo transcripts arising from ancestral, nongenic sequences are generally shorter and have fewer exons[20], thus showing reduced transcriptional cost compared to evolutionarily older transcripts[21], and which are likely to spread/fix under natural selection[22]. Notably, compared to PCGs (~2.34 isoforms per model) and TUCPs (~2.27 isoforms), lncRNAs exhibit analogous canonical splicing junction sites (Supplementary Fig. 7f) but relatively infrequent events of alternative splicing (~1.54 isoforms) (Supplementary Fig. 7d–e). The short circRNAs (~577 bp in length) tend to originate from linear, middle exons[23] (Supplementary Fig. 8c), and often contain multiple, canonical, linear exons (~172 bp) (Supplementary Fig. 8d–f) that have longer flanking introns (5375 bp vs. 1592 bp in the control) (Supplementary Fig. 8g).

We observed variation between the proportions of distinct transcript types that were transcribed across tissues. For example, compared to the smaller fractions of TUCPs (~36.94%), lncRNAs (~38.21%), miRNAs (~30.79%), and circRNAs (~9.37%) that showed evidence of transcription for a given tissue, more than two-third of PCGs (~76.66%) also presented evidence of transcription, ranging from ~65.65% in PIECs to ~89.38% in the testis (Fig. 1b), and with reduced tissue specificity in their expression (35.68% of PCGs with $\tau \geq 0.75$ vs. 70.70, 66.98, 69.24, and 95.43% for TUCPs, lncRNAs, miRNAs, and circRNAs, respectively)[12] (Fig. 1f). Remarkably, the testis, which has permissive chromatin that allows transcription of extensive genomic regions[24], had the highest proportion of transcribed PCGs (~89.38% vs. ~76.26% in each of the other tissues), TUCPs (~78.88% vs. ~35.63%), and lncRNAs (~76.10% vs. ~37.02%), but not the highest proportions of biogenesis-distinct short transcripts, i.e., circRNAs (~11.35% vs. ~9.31%) and miRNAs (~28.92% vs. ~30.85%) (Fig. 1b).

As expected, non-linear circRNAs were more abundant in brain tissues (23.27% and 11.67% were transcribed in the cerebellum and cerebrum, respectively, vs. 7.97% in each of the other tissues) (Fig. 1b). This finding was consistent with the notion that circRNAs are more likely to be derived from neuron-specific PCGs[16] and that their formation is enhanced by neuron-specific RNA-binding proteins[25] (e.g., quaking protein [QKI], whose transcript abundances were ~230.09 and 284.20 TPM in the cerebellum and cerebrum, respectively, vs. ~75.72 TPM in each of the other tissues). Moreover, a larger proportion of miRNAs were detected in the cerebrum (40.56%) and cerebellum (37.71%) than in each of the other tissues (30.26%) (Fig. 1b). This observation is consistent with miRNA-target coevolution, in which PCGs that are predominantly expressed in the brain generally have longer 3′ untranslated regions and thus are more likely to be targeted by miRNAs[26].

We also observed dissimilarities between the distribution of the abundances of distinct types of transcripts (i.e., complexity) across tissues. The most abundant transcripts (the top 0.5%, as ranked by expression levels) in a tissue, accounted for greater than half of the total transcribed TUCPs (~58.35%), lncRNAs (~75.53%), and miRNAs (~62.34%), whereas PCGs (~46.26%) and circRNAs (~35.48%) had a more uniform distribution (Fig. 1g and Supplementary Fig. 9). Among the few highly

**Table 1 The expanded pig transcriptome.**

| Transcripts | Detected transcripts | Annotated in the reference pig assembly (Sscrofa11.1)[a] | Features | | | | |
|---|---|---|---|---|---|---|---|
| | | | Length (bp) | Exon number | Isoform number | Tissue specificity (τ)[b] | Complexity[c] (%) |
| PCG | 20,505 | 21,303 | 3259 | 10.21 | 2.34 | 0.61 | 46.26 |
| TUCP | 2440 | \ | 3177 | 3.54 | 2.27 | 0.87 | 58.35 |
| LncRNA | 19,072 | 6797 | 1694 | 2.63 | 1.54 | 0.85 | 75.53 |
| circRNA | 48,232 | \ | 577 | 4 | \ | 0.96 | 35.48 |
| miRNA | 1245 | 462 (annotated in miRbase for pigs) | 23 | \ | \ | 0.90 | 62.34 |

A total of 867 miRNA precursors corresponded to 1245 mature miRNAs, including 251 known and 616 conserved precursors corresponding to 462 and 783 mature miRNAs, respectively. LncRNAs were classified into 14 different locus biotypes by location with respect to PCGs (Supplementary Fig. 5).
[a]Annotation release 102 of the Sscrofa11.1 assembly.
[b]Tissue specificity was calculated as the tau score (τ) (see Methods for details).
[c]Complexity is determined by the fraction of transcripts with the highest abundance (top 0.5%) accounting for the total number of transcribed transcripts for a tissue.

abundant PCGs (the top 0.5%; ~102) in a tissue, ~69.61% (~71) were ubiquitously transcribed across tissues ($\tau \leq 0.30$) and were commonly enriched in basic cellular functions, including "oxidative phosphorylation" ($P < 2.2 \times 10^{-16}$) and "regulation of translation" ($P < 2.21 \times 10^{-6}$). Nonetheless, ~7.84% of PCGs (~8) were specifically transcribed in a given tissue ($\tau \geq 0.75$). The categories "muscle contraction" ($P = 3.12 \times 10^{-4}$), "spermatogenesis" ($P = 3.37 \times 10^{-6}$), and "visual perception" ($P = 6.17 \times 10^{-7}$) were overrepresented among the highly abundant PCGs that were uniquely transcribed in SMTs (6 of 8 SMTs), the testis, and the retina, respectively (Supplementary Fig. 10).

Although distinct transcript types exhibited considerable differences in structural and expression characteristics, their transcriptional profiles were generally similar among tissues (Fig. 1c, d and Supplementary Fig. 11a), with circRNAs being more discriminative among tissues (Supplementary Fig. 11b). Testis clustered separately, while neural tissues (cerebrum, cerebellum, and retina) clustered together, whereas other functionally related tissues also co-clustered. Twelve striated muscles (eight SMTs and the four chambers of the heart), six ATs, and three digestive tissues (colon, small intestine, and stomach) also clustered into obviously separate respective groups (Fig. 1c and Supplementary Fig. 11a).

Supporting the notion that the transcriptional programs may store memory of lineage specification and differentiation[27], we observed that the transcriptomic divergence was relatively higher between tissues of different germ layers than within germ layers (Supplementary Fig. 12a). We observed that this profound difference was further amplified for 36 paralogous transcription factors (TFs) in the homeobox (HOX) superfamily, which are major regulators of animal morphogenesis and development (Supplementary Fig. 12b). We identified sets of germ layer-specific markers (2507 for ectoderm, 1053 for endoderm, and 874 for mesoderm) (Supplementary Fig. 12c) that were critical specifiers for different germ layers. For example, neurodevelopmental genes (such as HK1[28], VWC2[29], and NECTIN1[30]), a metabolic gene (FOXA3)[31], and genes in involved in cardiovascular differentiation (TMEM88[32] and VIM[33]) were highly expressed in tissues derived from ectoderm (i.e., cerebrum, cerebellum, and retina), endoderm (i.e., liver, small intestine, colon, and stomach), and mesoderm (typically, heart), respectively (Supplementary Fig. 12d).

**Visualization of transcriptional features in the nuclear space.**
In addition to the sequence of the genome, its three-dimensional (3D) structure plays a key role in regulating transcription[34]. To visualize the origins of transcription within the nuclear space, we reconstructed 3D genomic structures for subcutaneous AT (SAT) (as a representative somatic tissue) (Fig. 2) using an in situ high-throughput chromatin conformation capture (Hi-C) map of six pigs (a total of ~2.07 billion uniquely aligned contacts with the depth of ~344.29 million [M] contacts per library) (Supplementary Table 1 and Supplementary Data 4; see Supplementary Methods for details).

These 3D structures initially revealed the radial locations of chromosomes within the nuclear space. Visual examination showed that GC-rich chromosomes, which are usually transcript-rich (e.g., chromosome 17 has a 44.56% GC content and harbors ~10 PCGs per Mb), were preferentially found in the interior of the nucleus, while typically transcript-sparse GC-poor chromosomes (e.g., chromosome 1 with a 40.20% GC content harbors ~6 PCGs per Mb) preferentially localized in the nuclear periphery (Spearman's $r = -0.67$, $P = 2.45 \times 10^{-3}$ for GC content and $r = -0.51$, $P = 3.21 \times 10^{-2}$ for PCG density, when compared against the average radial position of each autosome with respect to the nuclear center) (Fig. 2a–c). We also observed correlations between primary transcription features and chromatin compartments (markers of chromatin accessibility). Compartment A (1105 Mb in length) regions were GC-rich (43.71%), transcript-rich (13.05 PCGs per Mb) and actively transcribed (5.04 TPM for PCGs), and were generally biased toward the center of the nucleus. In contrast, compartment B (1161 Mb in length) regions were GC-poor (38.26%), transcript-sparse (4.22 PCGs per Mb) and poorly transcribed (0.33 TPM for PCGs), and also tended to be localized in the periphery (Spearman's $r = -0.9$, $P < 2.2 \times 10^{-16}$, between the compartment A proportion and radial position with respect to the nuclear center) (Fig. 2d–f and Supplementary Fig. 13a–c).

At a finer scale, we found that PCGs within the same TAD, which were nearby in the genome and spatially closed, tended to be co-expressed across tissues more than the global average (binomial test, $P < 2.2 \times 10^{-16}$). These observations thus supported the involvement of topologically associating domains (TADs) in transcriptional regulation via restriction of chromatin interactions for regulatory sequences[35] (Supplementary Fig. 13d). The lncRNAs and TUCPs also exhibited similar patterns between transcription and genome folding (Fig. 2, Supplementary Fig. 13).

**Extraordinary diversity of transcription among Skeletal Muscle Tissues (SMTs).** SMTs comprise the largest tissues (by weight) in the pig body (30–70% of porcine carcass weight[36]) (Supplementary Fig. 14) and represent its most economically valuable products. Historically, research on SMTs has largely treated them as a single group of virtually interchangeable tissues. However, SMTs have highly diverse origins, shapes, metabolic features, and physical functions. In order to systematically survey and resolve the transcriptomic differences among these distinct SMT types,

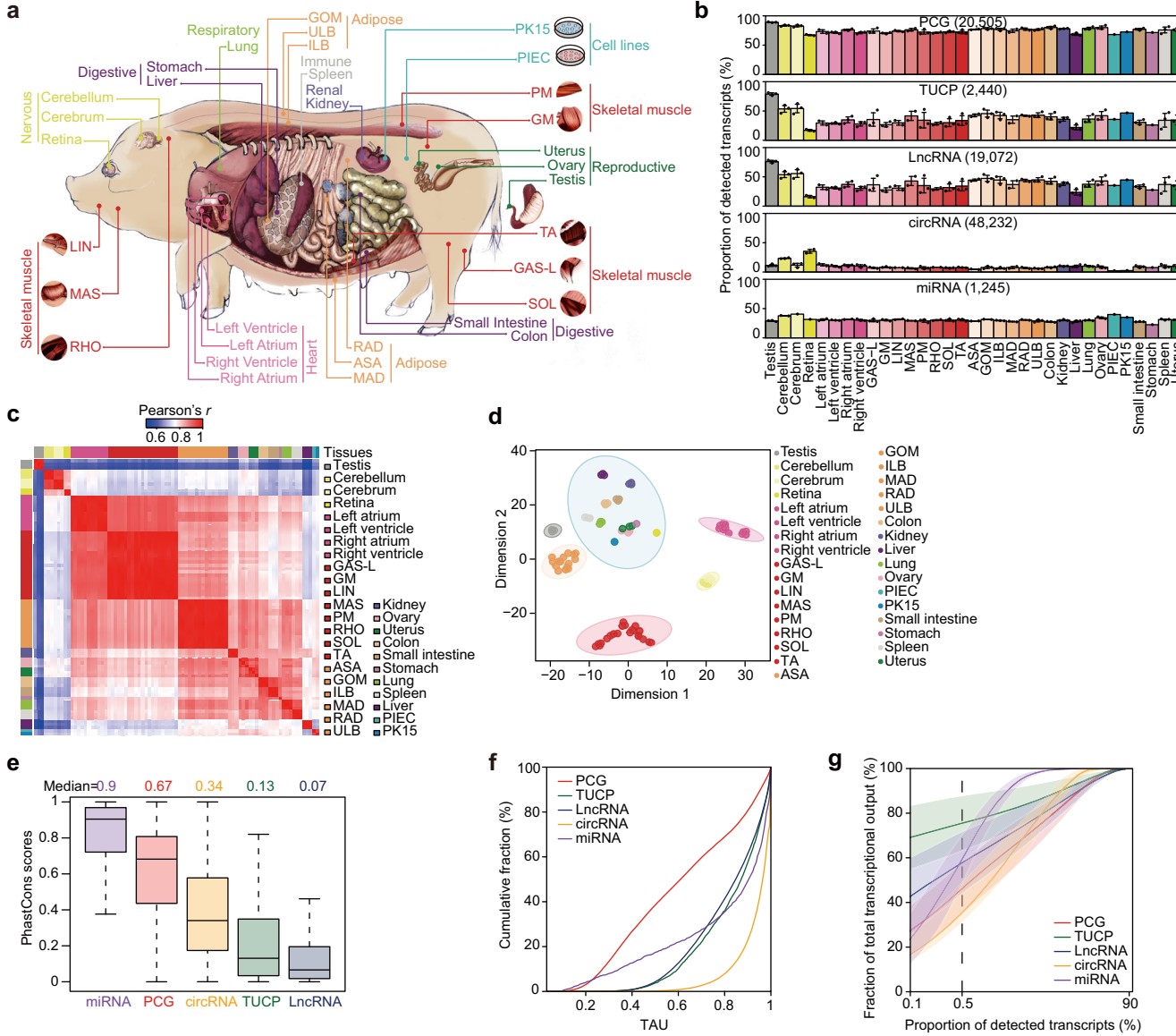

**Fig. 1 Characteristics of the pig BodyMap transcriptome. a** Samples derived from 31 tissues and two cell lines were used for ab initio pig transcriptome reconstruction. ASA: abdominal subcutaneous adipose, ILB: inner layer of backfat, ULB: upper layer of backfat, GOM: greater omentum, MAD: mesenteric adipose, RAD: retroperitoneal adipose, LIN: lingualis muscle, MAS: masseter muscle, RHO: rhomboideus muscle, TA: transversus abdominis muscle, GAS-L: gastrocnemius-lateralis muscle, GM: gluteus medius muscle, PM: *psoas* major muscle, and SOL: soleus muscle. See Supplementary Data 1 for details. **b** Proportions of detected transcripts across tissues and cell lines. The number in parentheses indicates the total number of transcripts with evidence of transcription in at least one sample (i.e., ≥1 TPM [transcripts per million] for miRNAs; ≥0.1 TPM for PCGs, TUCPs, and lncRNAs; and ≥0.05 TPM for circRNAs). Data are presented as mean values ± SD. **c, d** Hierarchical clustering (**c**) and t-distributed stochastic neighbor embedding (t-SNE) clustering (**d**) of samples using PCG expression. For the t-SNE plot, the ellipses indicate the tissues/cell lines with similar transcriptional profiles, constructed at a probability of 0.95. Notably, PK15 and PIEC cell lines were more similar to the epithelial- and endothelium-rich internal tissues (typically, ovary and uterus) than to nervous tissue, muscles, and adipose tissues. **e** Sequence conservation of five transcript types in the pig transcriptome (miRNA n = 321, PCG n = 19,041, circRNA n = 7673, TUCP n = 1119, lncRNA n = 8678). The base (nucleotide resolution) phastCons scores were collected from the UCSC Genome Browser based on Multiz alignment of 100 vertebrate species. Only pig sequences that could be matched to human sequences were used. The transcript-level phastCons scores were calculated as their average value among exonic sequences. In the boxplot, the internal line indicates the median, the box limits indicate the upper and lower quartiles and the whiskers extend to 1.5 IQR from the quartiles. **f** Tissue specificity of different types of transcripts reflected by tau (τ) score. **g** Abundance distribution of distinct types of transcripts across tissues and cell lines. The x-axis indicates the proportion of transcripts sorted from highest to lowest abundance, with the vertical dashed line indicating the top 0.5% of highest abundance transcripts. The y-axis indicates the accumulated fraction of transcripts relative to the total transcripts. Colored lines represent mean values across tissues, and lighter-colored shading around the mean represents dispersion calculated using the standard deviation divided by the cumulative sum of all means. Source data for **b**–**g** are provided as a Source Data file.

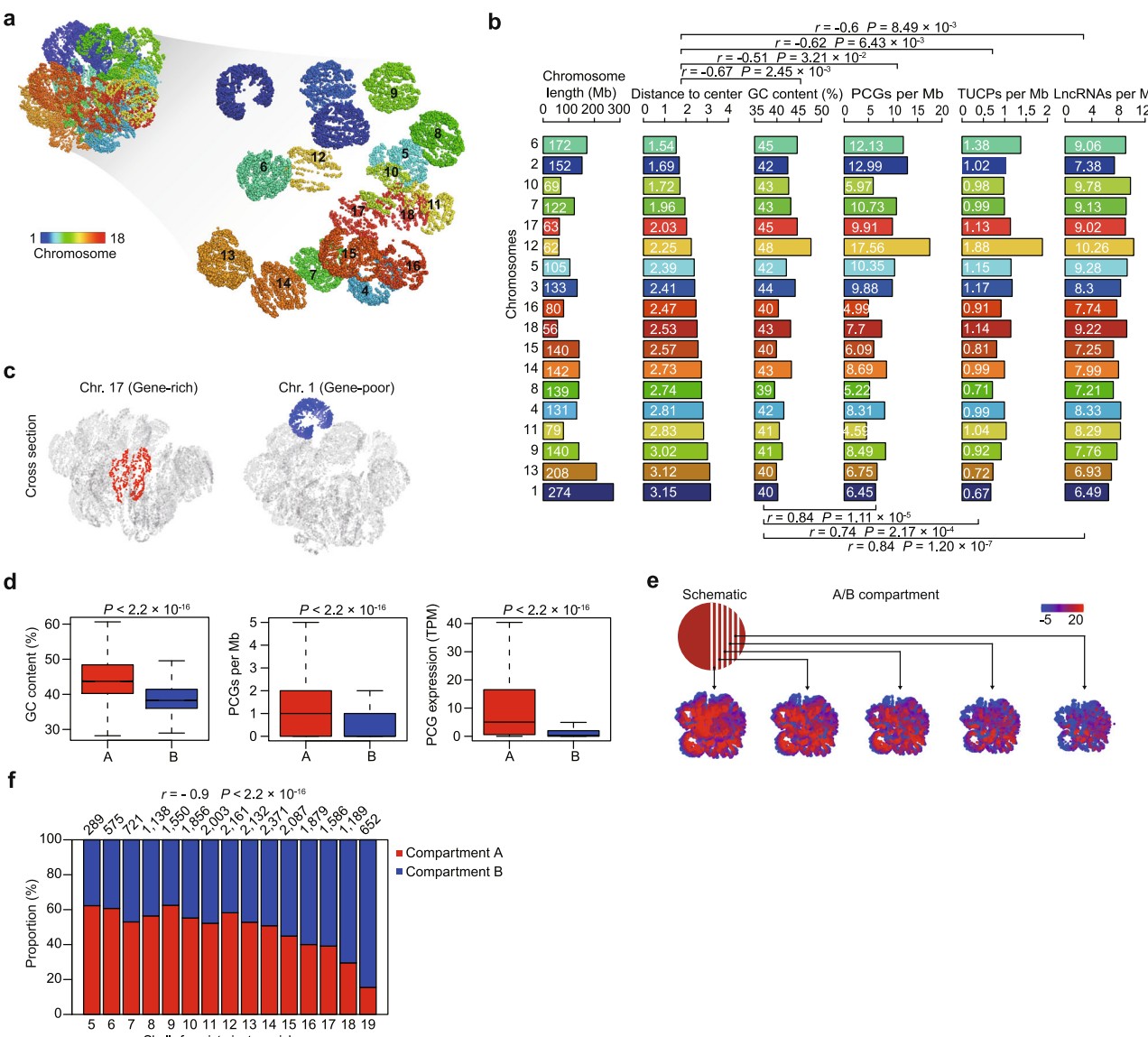

**Fig. 2 Transcriptional features in the nuclear space. a** 3D genome structure of the representative subcutaneous AT. Left: whole genome; right: separate chromosomes. X chromosome is excluded from this analysis due to the overriding effect of X-chromosome inactivation. Chromosomes are distinguished by color. Each point represents 100 kb of chromatin. **b** Bar plot indicating chromosome distances to the nuclear center and sequence features of each chromosome. There was a positive correlation between GC content and gene density for PCGs ($r = 0.84$), lncRNAs ($r = 0.74$), and TUCPs ($r = 0.84$). There was a negative correlation between the distance to the center of nucleus and features, including GC content ($r = -0.67$), PCG density ($r = -0.51$), lncRNA density ($r = -0.62$), and TUCP density ($r = -0.60$). The statistical significance of the two-sided $P$ value was calculated using hypothesis testing. The chromosomes are ranked by distance from the interior (upper) to the periphery (lower). **c** Examples of preferential localization in the nucleus for two chromosomes. Gene-rich chromosome 17 preferentially localized to the nuclear interior (left), whereas gene-poor chromosome 1 was consistently observed on the nuclear periphery (right). **d** Comparison of genomic properties including GC content, PCG density, and PCG expression between compartments A (1105 Mb in length) and B (1161 Mb in length). ($P$ values determined by two-sided Wilcoxon test). In the boxplot, the internal line indicates the median, the box limits indicate the upper and lower quartiles, and the whiskers extend to 1.5 IQR from the quartiles. GC content: $n = 11,060$ (A), $n = 11,608$ (B); PCGs per Mb: $n = 11,060$ (A), $n = 11,608$ (B); PCG expression: $n = 14,430$ (A), $n = 4898$ (B). **e** 3D models of the pig genome in SAT. Compartments A/B (at 100 kb resolution) were aggregated in 3D space. The plot was visualized in quintuplicate, with five intersecting sections plotted from the interior regions of the nucleus (left) to the periphery (right) based on distance (schematically depicted in the upper-left inset). Color bar indicated the levels of compartments A/B. **f** Percentage graph showing trends in the arrangement of the A/B compartments from the interior to the periphery within the nucleus. The nucleus is equally divided into 20 shells based on the distance to the 3D nuclear center. The numerical value above each bar indicates the bin (20 kb in size) number within this shell. There was a negative correlation (Spearman's $r = -0.9$, $P < 2.2 \times 10^{-16}$) between the compartment A proportion and the relative distance to the center of the nucleus. The statistical significance of the two-sided $P$ value was calculated using hypothesis testing. Shells with fewer than 250 bins were excluded from this analysis and are not shown. Source data are provided as a Source Data file.

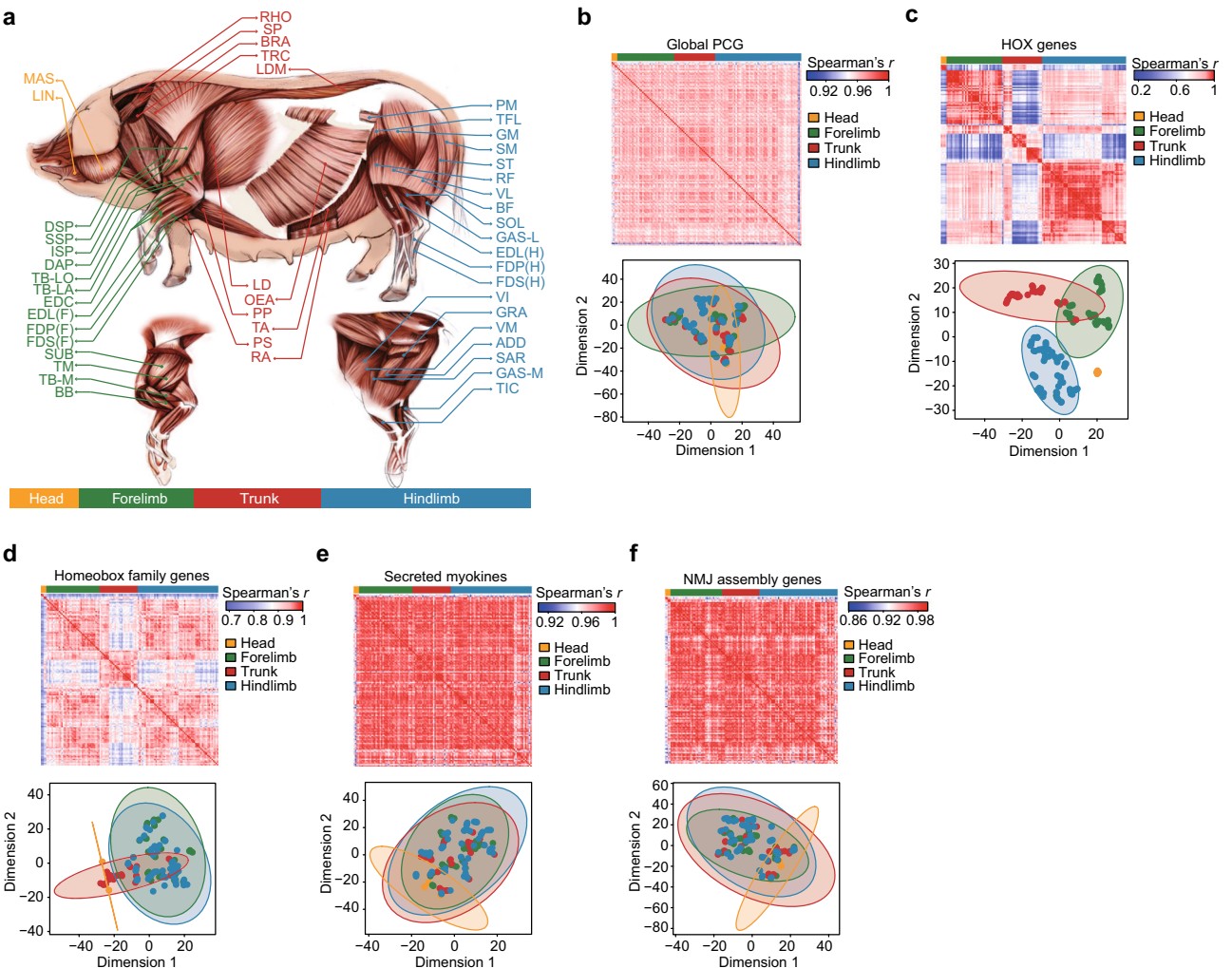

**Fig. 3 The transcriptional diversity among SMTs. a** Illustration of the anatomical position of SMTs. **a** A total of 47 types of SMTs were used to compare transcriptional profiles between anatomical regions, classified into four general groups: two head SMTs (orange) (LIN: Lingualis, MAS: Masseter); 14 forelimb SMTs (green) (BB: Biceps brachii, DAP: Deltoideus acromial part, DSP: Deltoideus scapular part, EDC: Extensor digitorum communis, EDL(F): Extensor digitorum lateralis (forelimb), FDP(F): Flexor digitorum profundus (forelimb), FDS(F): Flexor digitorum superficialis (forelimb), ISP: Infraspinatus, SSP: Supraspinatus, SUB: Subscapularis, TB-LA: Triceps brachii caput lateralis, TB-LO: Triceps brachii caput longum, TB-M: Triceps brachii caput medialis, TM: Teres major); 11 trunk SMTs (red) (BRA: Brachiocephalicus, LD: Latissimus dorsi, LDM: Longissimus dorsi muscle, OEA: Obliquus extensor abdominis, PP: Pectoralis profundus, PS: Pectoralis superficialis, RA: Rectus abdominis, RHO: Rhomboideus, SP: Splenius, TA: Transversus abdominis, TRC: Trapezius cervicis); and 20 hindlimb SMTs (blue) (ADD: Adductores, BF: Biceps femoris, EDL(H): Extensor digitorum lateralis (hindlimb), FDP(H): Flexor digitorum profundus (hindlimb), FDS(H): Flexor digitorum superficialis (hindlimb), GSA-L: Gastrocnemius-lateralis, GAS-M: Gastrocnemius-medialis, GM: Gluteus medius, GRA: Gracillis, PM: Psoas major, RF: Rectus femoris, SAR: Sartorius, SM: Semimembranosus, SOL: Soleus, ST: Semitendinosus, TFL: Tensor fasciae latae, TIC: Tibialis cranialis, VI: Vastus intermedius, VL: Vastus lateralis, VM: Vastus medialis). **b–f** Similarity of global PCG transcription (**b**) and transcriptional profiles between SMTs from different anatomical regions for *HOX* genes (**c**), homeobox family genes (**d**), secreted myokines (**e**), and neuromuscular junction (NMJ) assembly genes (**f**) based on Spearman's correlation (upper panel) and multidimensional scaling distances determined by t-SNE (lower panel). Ellipses in the t-SNE plots indicate the four SMT groups, constructed at a probability of 0.95. Source data for **b–f** are provided as a Source Data file.

we analyzed 130 RNA-seq datasets and their 130 corresponding small RNA-seq datasets from 47 SMTs (1–3 libraries for each SMT), which were derived from four anatomically distinct regions throughout the body (two in the head, 14 in the forelimb, 11 in the trunk, and 20 in the hindlimb) (Fig. 3a). Consistent with our findings from the core atlas dataset from 31 tissues and two cell lines (Fig. 1), we identified intrinsic differences in the transcriptional profiles of different transcript types (Supplementary Fig. 15). We observed extensive transcriptional heterogeneity among different SMTs. For example, the transcribed proportion of PCGs across SMTs ranged from 67.40% in the vastus lateralis (VL) to 79.75% in the hindlimb flexor digitorum profundus

(FDP-H) (Supplementary Fig. 15a), which was a comparable difference to that observed between the metabolically active liver (68.52%) and the stimulated cerebrum (82.64%).

To investigate the potential developmental history of anatomically distinct SMTs, we checked the expression of paralogous transcription factors in the *HOX* superfamily, which are major regulators of animal morphogenesis and development with respect to formation of the anterior-posterior axis (i.e., from head to hindlimb)[37,38]. A large proportion of *HOX* paralogs (35 of 37, or 94.59%) had evidence of transcription in SMTs (TPM ≥0.1 in at least one SMT). Multidimensional scaling analysis revealed that these *HOX* genes exhibited a clear pattern of

independent clustering based on their differences in expression among these four anatomically distinct regions (Fig. 3c). Notably, anatomically symmetrical SMTs (e.g., four pairs of SMTs in the forelimb and hindlimb) were clustered together with their respective counterparts. Furthermore, anatomically neighboring SMTs generally exhibited more similar patterns of expression than those from arbitrarily paired anatomical regions from head to hindlimb. For example, four SMTs (brachiocephalicus [BRA], splenius [SP], rhomboideus [RHO], and trapezius cervicis [TRC]) that located around the neck in the front trunk, were clustered with two SMTs (lingualis [LIN] and masseter [MAS]) in the head (Supplementary Fig. 16a). The expression profiles of 129 (out of 149, or 86.58%) homeobox family members transcribed in SMTs reiterated these findings (Fig. 3d, Supplementary Fig. 16b) and further revealed several master regulators that function in specific SMTs. For example, *MEOX2*[39], *PRRX1*[40], and *LHX2*[41] transcripts, which are the essential components regulating head muscle patterning, were more abundant in masseter in the head than in other SMTs (Supplementary Fig. 16c).

SMTs influence physiology and activity throughout the body by secreting numerous functionally essential hormones (myokines) involved in autocrine regulation of metabolism[42]. The vast majority of myokines (551 of 555, or 99.28%) showed evidence of transcription in at least one SMT, of which nearly half (271 of 551, or 49.18%) were ubiquitously transcribed ($\tau \leq 0.3$). Remarkably, 222 myokines were highly abundant (TPM >10) in almost all SMTs, reflecting their essential activities in SMTs, including angiogenesis (such as *VEGF*), NAD biosynthesis (such as *NAMPT*), and inflammatory response (such as *AIMP1, CMTM6, CXCL12*, and *HMGB1*)[42–45]. Only a few myokines exhibited differences in abundance between SMTs (~19 of 551, or 3.45%, in pairwise comparisons between SMTs), which suggested that the expression profiles of myokines could be mostly indistinguishable among different SMTs (Fig. 3e and Supplementary Fig. 17)[42,46].

SMTs are voluntarily controlled by the nervous system through the formation of appropriate neuromuscular junction (NMJ) assemblies. Understanding how NMJ-related PCGs are expressed across SMTs is therefore fundamental to neurobiology and regenerative medicine[47,48]. We detected the transcripts of 77 (of 77, or 100%) NMJ-related PCGs in at least one SMT. A considerable fraction (26 of 77, or 33.77%) were ubiquitously expressed ($\tau \leq 0.3$). Among these PCGs, 19 were involved in synapse formation and were highly abundant (TPM >10) in almost all SMTs (Fig. 3f and Supplementary Fig. 18a). These results suggested that they were bona fide SMT-derived transcripts with a conserved role in NMJ assembly, and therefore excluded the possibility of contamination from nearby neurons. For example, *DVL1* (TPM = 37.52) encodes a signaling protein necessary for guiding axons to the middle region of myofibers to form NMJs[49], and the disruption of which causes synaptic defects at the NMJ[50]. *APP*, which encodes a key regulator of neuromuscular synapse structure and function, was widely present in synapses of the NMJ (TPM = 22.6). *APP* deficiency leads to a dramatic reduction in presynaptic terminals and aberrant synaptic failure in mice[51]. Only a few NMJ-related PCGs (~2 of 77, or 2.6%) showed differences in their abundance in pairwise comparisons between SMTs, which implied intrinsic similarities in forming neuromuscular synapses and generating force among SMTs. Notably, nine NMJ-related PCGs (13 of 77, or 16.88%) differed in abundance between SMTs in the head versus other anatomical regions. *SIX1*, a transcription factor essential for the development of trigeminal ganglia[52], is highly expressed in MAS in the head (TPM = 44.25 vs. 16.1 in other SMTs), which is innervated by a branch of trigeminal ganglia (Supplementary Fig. 18b).

**Myofiber types are associated with transcriptomic divergence among SMTs.** Mammalian SMTs are mainly composed of heterogeneous myofibers with marked differences in contractile and metabolic features and can thus be classified into different types based on the transcription of their respective myosin heavy chain (*MYH*) isoforms. We performed clustering analysis to compare expression patterns of ten annotated *MYH* isoforms across SMTs (Supplementary Fig. 19a) and found that they were weakly consistent with the global transcription patterns of PCGs and other transcript types (cophenetic $r = 0.343$ for PCGs, 0.328 for TUCPs, 0.329 for lncRNAs, 0.060 for circRNAs, and 0.138 for miRNAs). This result suggested that the composition of myofiber types determined by *MYH* isoforms was insufficient to establish the diversity of transcriptional patterns across SMTs[46].

We next focused on three of four dominant myofiber types (types I, IIA, and IIB, respectively, correspond to *MYH7*, *MYH2*, and *MYH4* isoforms), whereas the *MYH1* isoform for type IIX was not quantified due to mis-annotations in the current reference genome assembly. We found that changes in the abundance of the type I, IIA, and IIB marker isoforms were positively correlated with global divergence in the transcriptome (reflected by the proportions of each of the five transcript types that showed altered expression in pair-wise comparisons of SMTs) (Supplementary Fig. 19b–f). This result suggested that differences in the physical properties between myofiber types may be associated with transcriptomic divergence among SMTs containing different myofiber composition. Highly specialized oxidative type I myofiber (e.g., *MYH7*), which provides low force output but long endurance (Pearson's $r = 0.27$, $P < 2.2 \times 10^{-16}$) and glycolytic type IIB myofiber (e.g., *MYH4*), which enables the highest output of force but susceptibility to fatigue (Pearson's $r = 0.41$, $P < 2.2 \times 10^{-16}$) may both more profoundly influence the transcriptomic divergence among SMTs than type IIA myofiber (e.g., *MYH2*), which has intermediate features between those of types I and IIB (Pearson's $r = 0.16$, $P = 5.13 \times 10^{-7}$).

In addition, we further observed that myofiber-specific PCGs (i.e., the top 10% of PCGs with the highest Pearson's $r$ for the expression of their respective *MYH* isoforms across SMTs) were overrepresented in different GO enrichment categories between type I (*MYH7*) and IIB (*MYH4*) myofibers (Supplementary Fig. 20), further supporting the influence of these isoforms on transcriptomic divergence. The 432 type I myofiber-specific associated PCGs were mainly enriched in categories related to lipid metabolism, such as 'fatty acid catabolic process' (44 PCGs, $P < 2.2 \times 10^{-16}$) and 'lipid biosynthetic process' (37 PCGs, $P = 8.98 \times 10^{-10}$) (Supplementary Fig. 20b). This finding confirmed published observations that the triacylglycerol content and fatty droplets were higher in type I compared to type IIB fibers[53]. For example, *CPT2*, encoding an essential nuclear protein in mitochondrial long-chain fatty acid oxidation, was specifically associated with type I myofibers (Pearson's $r = 0.72$ with *MYH7*, $P < 2.2 \times 10^{-16}$). By contrast, the 89 type IIB myofiber-specific associated PCGs were mainly involved in 'glycolytic process' (15 PCGs, $P < 2.2 \times 10^{-16}$) and 'muscle contraction' (15 PCGs, $P = 6.12 \times 10^{-13}$), potentially reflecting type IIB myofiber demands for rapid contractile speeds and the need to generate energy via anaerobic glycolytic metabolism during short-term activity[54]. For example, we found that a negative regulator of skeletal muscle cell proliferation and differentiation, *MSTN*, was specifically associated with type IIB myofibers ($r = 0.75$ with *MYH4*, $P < 2.2 \times 10^{-16}$). This observation was consistent with the especially strong inhibitory role of *MSTN* in type IIB myofibers compared to that in type I[55]. Notably, loss of *MSTN* function has been associated with markedly increased proportions of type IIB myofibers but no change in, or even reduced numbers of, type I myofibers in mice[56].

To identify potential TUCP, lncRNA, circRNA, and miRNAs potentially involved in specialized muscle function, we also separately identified putative regulatory transcripts that were co-expressed with *MYH* markers for type I, IIA, and IIB myofibers (Pearson's $r > 0.4$, $P < 0.05$; or the top 10%) by checking for functional enrichment of their *cis*-regulated neighboring PCGs (100 kb up- and downstream) or PCG targets of miRNA (Supplementary Fig. 20c–e and Supplementary Data 5).

**Myofiber type composition of SMTs**. To accurately depict the myofiber composition of SMTs in pigs, we used a single-cell resolution spatial transcriptomics (ST) approach to dissect the transcriptional differences among type I, IIA, and IIB myofibers of the psoas major (PM) muscle as a representative SMT (Fig. 4, Supplementary Table 2). We subsequently identified several hundred myofiber-specific PCGs, and detected differentially activated metabolic and signaling pathways in specific myofibers, thus leading to more defined PCG signatures (~446) for three highly specialized myofibers (Supplementary Figs. 21–24). Based on the signatures of these myofiber-specific PCGs, we next used a CIBERSORTx deconvolution method[57] to estimate the compositions of type I, IIA, and IIB myofibers (indicated by the proportion of each myofiber type among the total) in 130 bulk RNA-seq datasets from 47 SMTs (Supplementary Data 6; Supplementary Methods). Compared with estimates of the proportion of type I myofibers reflected by a single marker (*MYH7*), we found that the assembly of myofiber-specific PCG signatures provided an estimate of the type 1 proportion that was more consistent with measurements of ATPase staining across 30 SMTs (Spearman's $r = 0.26$ with $P = 0.16$ for the *MYH7* marker vs. $r = 0.47$ with $P = 9.42 \times 10^{-3}$ for multiple PCG signatures) (Supplementary Fig. 25). As ST analyses simultaneously resolve spatial patterns at the single myofiber- and transcriptome-wide scales, they enable a much more detailed view of myofibers and result in much greater discrimination power for different myofibers.

Divergence of myofiber composition between SMTs from different anatomical regions reflected their distinct physiological properties (Fig. 4e). For instance, we found that relatively slow soleus (SOL) (fatigue-resistant muscle involved in posture) had a larger proportion of type I (53.12%) and fewer type IIA (46.71%) and IIB (0.16%) myofibers than the fast extensor digitorum lateralis in the hindlimb (EDL-H) (26.00% for I, 49.05% for IIA, 24.95% for IIB; fatigable muscle involved in rapid movement) (Fig. 4e), which aligned with patterns observed in humans and mice[58]. Although the masseter and tongue have similar developmental histories, they have distinctly different myofiber compositions. Compared to the masseter, which contains a higher proportion of type I (61.61%) but fewer type IIA (38.39%) and no IIB myofibers, the tongue has fewer type I (36.82%) but more type IIA (63.19%) (and also no type IIB) myofibers (Fig. 4e). Intriguingly, the predominance of type I myofibers in the pig masseter resembles that in the human masseter[59] but differs from that in the mouse masseter (~75% type IIB but lacking type I)[46]. This finding supports the notion that myofiber composition can substantially differ between analogous SMTs among mammals. Even between the evolutionarily close mice and rats, surprising differences have been reported[46].

Generally, although anatomically neighboring SMTs can be classified into a single group, they may have dissimilar myofiber composition and thus, distinct physical properties. These differences were evident in comparisons with the caput medialis of the triceps brachii (TB-M), which revealed that the comparatively superficial caput lateralis of TB (TB-LA) and caput longum of TB (TB-LO) tissues had fewer type I (19.57% for TB-LA and 24.28% for TB-LO vs. 45.69% for TB-M) and more type IIB (38.28% for TB-LA and 36.69% for TB-LO vs. 21.02% for TB-M) myofibers. Moreover, the pectoralis profundus (PP) (26.77% type I, 40.91% type IIA, and 32.32% type IIB) exhibited the opposite composition of its superficial counterpart (i.e., 6.99% type I, 24.66% type IIA, and 68.35% type IIB for the pectoralis superficialis [PS]). These results suggested that the deeper layer of SMTs (generally involved in maintaining posture) had a higher oxidative metabolism and were thus more likely to contain a higher proportion of type I myofibers than their superficial counterparts (involved in rapid movements), which contained more type IIB myofibers[60].

**Metabolic and inflammatory divergence between subcutaneous and visceral ATs**. Adipose tissues (ATs) can develop in multiple discrete locations in mammals, and have been treated as separate 'miniorgans' due to differences in their functional properties[61,62]. To investigate transcriptomic divergence potentially linked to morphological and functional heterogeneity among anatomically distinct ATs, we analyzed 15 RNA-seq datasets and their 14 corresponding small RNA-seq datasets from three subcutaneous ATs (SATs) and three visceral ATs (VATs), with 2–3 libraries for each AT type.

As expected, we found that the transcriptional profiles of each transcript type were generally distinguishable between SATs and VATs (Supplementary Fig. 26a). The PCGs upregulated in SATs (~196 in SAT vs. VAT comparisons) were mainly involved in categories related to "organization, structural constituent and space of extracellular matrix (ECM)" (Supplementary Fig. 26b). This result supported the metabolically protective roles of SATs because relaxation of the ECM allows excess nutrient storage in adipocytes and avoids the pathological features that include activation of stress-related pathways, inflammation, and ectopic lipid deposition in other tissues[63]. Nonetheless, the PCGs upregulated in VATs (~539 in SAT vs. VAT comparisons) were enriched in multiple categories related to "inflammation" and "immunity" (Supplementary Fig. 26b), which was consistent with reported increases in inflammatory and immune progenitors in VATs compared with SATs[63]. In contrast to the direct connection between SATs and systemic circulation, venous blood in VATs is drained directly to the liver, and numerous signaling and mediator proteins secreted by VATs can directly access hepatic cells[64], suggesting VATs carry higher metabolic risks than SATs.

AT is composed of a collection of heterogeneous cell types[65], and changes in the composition of cell subpopulations (in particular, a marked increase in immune cell infiltration) have been proposed to contribute some of the negative health consequences of obesity[63]. To investigate inflammatory characteristics of visceral greater omentum (GOM) adipose, we used in silico deconvolution analysis with CIBERSORTx[57] to estimate the relative proportions of four typical cell types in ATs (adipocytes, macrophages, CD4$^+$ T cells, and microvascular endothelial cells [MVECs]) in 15 bulk SAT transcriptomes of pigs using the orthologous cell-type-specific marker genes from their respective purified single-cell transcriptomes in humans[65] (Supplementary Methods). We then compared the proportions of these cell types in GOM with that in other VATs, including mesenteric adipose (MAD) and retroperitoneal adipose (RAD), and with three SATs, including abdominal subcutaneous adipose (ASA), upper layer of backfat (ULB), and inner layer of backfat (ILB), to highlight the stronger inflammatory response in GOM. Notably, we found that GOM exhibited higher infiltration of macrophages (M1/M2 combined) (9.81%) than the other AT tissues (MAD = 7.39%; RAD = 6.34%; ASA = 7.18%; ULB = 4.67%; ILB = 5.70%) (Supplementary Fig. 27). Indeed, when the three VATs were separately compared with each of the SATs, GOM had a higher average upregulated inflammatory PCGs

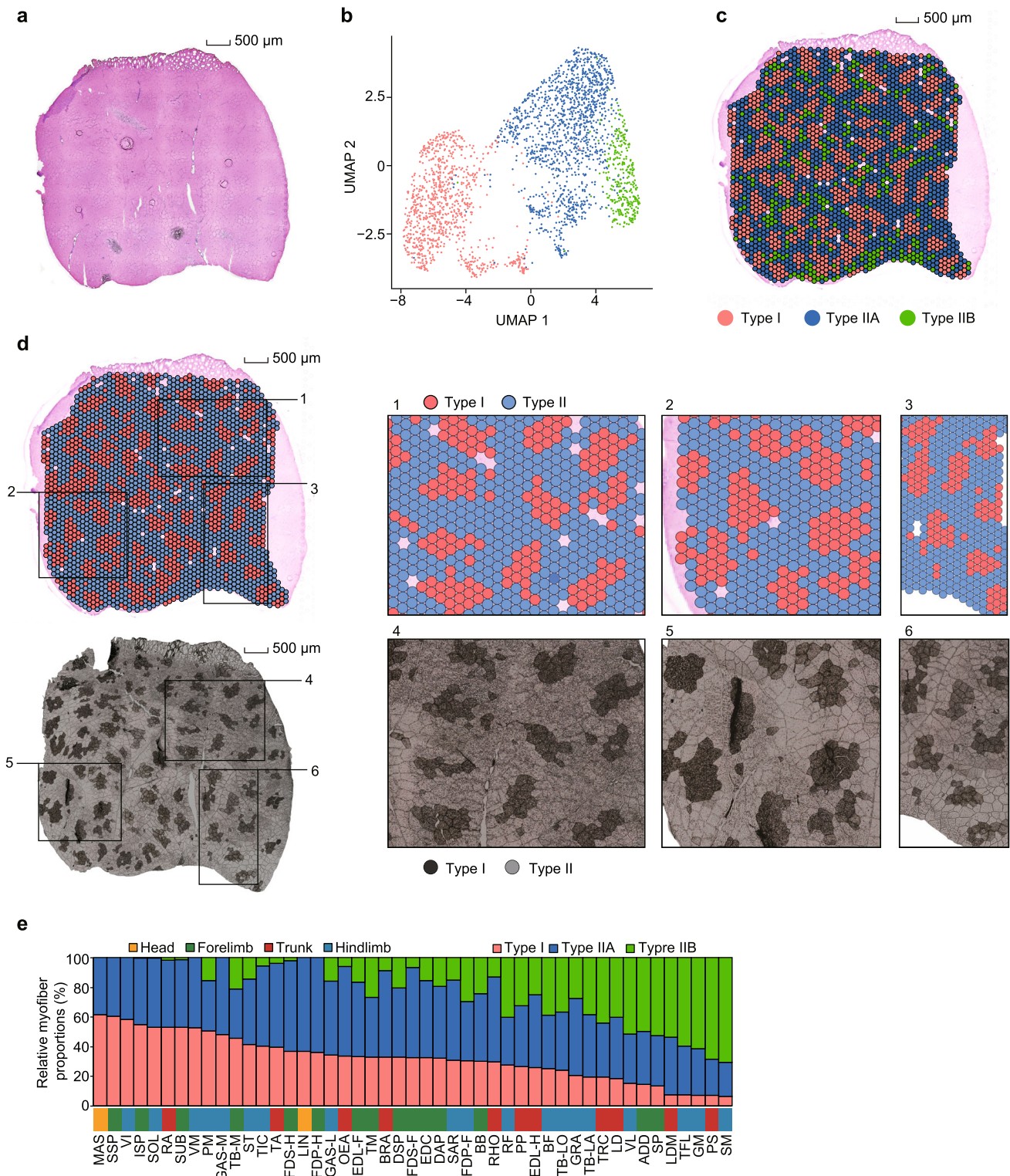

**Fig. 4 Spatial transcriptomics (ST) illustrate the myofiber composition of SMTs. a** Hematoxylin-and-eosin staining of psoas major (PM) muscle sections. Scale bars, 500 μm. A representative image of two independent experiments is shown. **b** Dimensionality reduction and clustering of 2607 spots (replicate 2) identified as type I, IIA, and IIB myofiber clusters. The identified putative myofiber/spot clusters are annotated based on *MYH* transcriptional profiles, as well as marker gene transcription, shown in detail in Supplementary Figs. 21 and 22. **c** Mapping of data points from three putative myofiber clusters to their spatial positions. **d** Comparison of the location of type I myofibers identified in ST (upper panel) and ATPase-stained histological sections (lower panel) for replicate 2 (tissue sections adjacent to that in **a**). In the ST plot (upper panel), type IIA and IIB myofibers were combined (shown in light blue). Scale bars, 500 μm. Insets show three typical regions/images (denoted with numbers) magnified for comparison matching results (1 vs. 4, 2 vs. 5, and 3 vs. 6). **e** Estimates of the myofiber proportions in bulk RNA-seq data using the expression signature matrix (442 and 450 genes for replicate 1 and replicate 2, respectively) of the three myofibers, shown as the average of replicate 1 and replicate 2. SMTs are ranked by their proportions of type I myofiber from high (left) to low (right). Source data are provided as a Source Data file.

(~48) than MAD (~31) and RAD (~7) (Supplementary Fig. 28a, c) but a greater average number of downregulated ECM-related PCGs (43) than either MAD (29) or RAD (15) (Supplementary Fig. 28b, d). In particular, the *ALOX15* and *F2R*, inflammatory markers[66,67] were specifically upregulated in GOM (Supplementary Fig. 28e). These results led us to postulate that the lower activity of ECM-related PCGs in GOM may lead to limited space in the adipocyte ECM and consequently result in metabolic deterioration of GOMs during expansion of ATs[63], thus supporting a metabolically harmful role of GOM.

In addition, we found that the transcription of 29 *HOX* paralogs in GOM and MAD were most dissimilar with that in three SATs, supporting the likelihood that SATs and VATs originate from different progenitor lines during development[68,69]. Strikingly, the visceral RAD clustered more closely with SATs and was originally distinguishable from the congeneric GOM and MAD (Supplementary Fig. 29a). The expression patterns of 95 (66.43% of 143) homeobox family PCGs across ATs also supported this relationship. Notably, *HOXC9* (an indicator of a beneficial metabolic phenotype and protector against obesity-related insulin resistance)[70] was upregulated in RAD (~3.28 TPM) and SATs (ASA: ~5.10 TPM; ULB: ~1.21 TPM; ILB: ~1.17 TPM) compared to GOM (~0.14 TPM) and MAD (~0.22 TPM) (Supplementary Fig. 29b, c).

As metabolic risk is often reflected by reduced responsiveness to catabolic and anabolic signals[71], we further examined the transcription of 22 PCGs essential for glucose and lipid metabolism[72] and found that RAD exhibited greater similarity to metabolically active SATs in its transcriptional patterns of these PCGs than to either GOM or MAD. In particular, PCGs involved in lipid and sterol synthesis (e.g., *ACACA*), lipolysis (e.g., *PNPLA2*), gluconeogenesis (e.g., *HDAC4*), glycolysis (e.g., *PFKFB3*), and glycogen uptake (e.g., *SLC2A4*, also named *GLUT4*) and storage (e.g., *GYS1*) were differentially upregulated in RAD and SATs compared to GOM and MAD. These results further supported similarities between RAD and SATs[73] (Supplementary Fig. 29d). Further cell lineage tracing is necessary to determine the full heterogeneity of progenitor lines across ATs and to better pinpoint the potential ontogenetic sources of RAD.

**Diverse mitochondrial pathways across tissues.** Mitochondria, which encode a total of 13 PCGs (mt-PCGs) that are all core components of oxidative phosphorylation (OXPHOS) in vertebrates[74], are semiautonomous organelles that house numerous biochemical pathways and perform central functions in apoptosis and ion homeostasis. Using 1043 nuclear PCG entries in MitoCarta2.0[75] that are annotated as post-translationally imported into mitochondria (mt-localized nu-PCGs) (Fig. 5a), we constructed a compendium of the mitochondrial transcriptome across 70 tissues and two cell lines.

All 13 mt-PCGs exhibited prominent housekeeping features (i.e., extremely high abundances, ~736.27 TPM; lower tissue specificity, $\tau =$ ~0.20). Compared with the nuclear-encoded PCGs outside the mitochondria ($\tau =$ ~0.70 and TPM = ~2.13), the mt-localized nu-PCGs were more ubiquitously transcribed and in higher abundance ($\tau =$ ~0.38 and TPM = ~10.78). Of 1043 mt-localized nu-PCGs, 918 (88.02%) were transcribed in at least one tissue/cell line, and 817 of which were detected in all tissues/cell lines (Fig. 5b). We identified 89 (9.69% of 918) mt-localized nu-PCGs that were transcribed only in a specific tissue ($\tau \geq 0.75$) (Fig. 5b) and were enriched in functional categories corresponding to known tissue-specific mitochondrial pathways (Supplementary Fig. 30). For example, three mt-localized nu-PCGs (*CPS1*, *NAGS*, and *OTC*) that encode enzymes involved in the urea cycle, an essential pathway for ammonia detoxification

restricted to periportal hepatocytes, were extremely abundant in the liver (*CPS1*: 1007.55 in TPM vs. ~2.61 in other tissues; *OTC*: 346.83 in TPM vs. ~1.14; *NAGS*: 43.88 in TPM vs. ~0.31). The transcription of 13 mt-PCGs and 918 mt-localized nu-PCGs revealed the general tissue diversity of mitochondrial pathways/inventories, especially for metabolically active striated muscles (including SMTs and the four chambers of the heart) and ATs, which clustered with their respective counterparts, but differed from other tissues/cell lines (Fig. 5c).

We next examined the mitochondrial transcriptome across tissues with respect to two central metabolic pathways of energy production, i.e., OXPHOS and fatty acid β oxidation (FAO). We found that more than half of the PCGs involved in OXPHOS (75 of 111, or 67.57%; $P < 2.2 \times 10^{-16}$, $\chi^2$ test) and FAO (36 of 68, or 52.86%, $P < 2.2 \times 10^{-16}$, $\chi^2$ test) were located in mitochondria, supporting the essential function of mitochondria in metabolism[76]. All of these genes (75 of 75 for OXPHOS; 36 of 36 for FAO) were detected in at least one tissue/cell line. As expected, to meet the higher metabolic cost for striated muscles during intermittent and continuous tetanic contractions, the SMTs and heart exhibited a more active OXPHOS system than other tissues (58 of 75, or 77.33%, of OXPHOS-related mt-localized PCGs were upregulated) (Fig. 5d and Supplementary Fig. 31). Supporting the marked oxidative capacity of highly specialized oxidative type I myofibers, the transcription of *MYH7* (type I marker) was highly correlated with OXPHOS-related mt-localized nu-PCGs across SMTs (average Pearson's r = 0.37 vs. 0.23 for type IIA *MYH2* and -0.07 for type IIB *MYH4*) (Supplementary Fig. 32). This association was even stronger than that between the mitochondrial oxidative metabolism *PGC-1α*- and OXPHOS-related mt-localized nu-PCGs (Pearson's r = 0.26)[77].

FAO in mitochondria is an alternative pathway for energy production. Accordingly, six ATs exhibited more distinct transcriptional profiles for the 36 FAO-related mt-localized nu-PCGs than other tissues (Fig. 5e). We observed that ATs specifically contained upregulated PCGs essential for FAO, such as *ACAA1* (peroxisomal β oxidation) (TPM = 83.08 vs. 7.30 for other tissues) and *ECHDC1* (metabolite proofreading) (TPM = 46.77 vs. 8.45) (Supplementary Fig. 33). Consistent with previous reports of a larger mtDNA copy number in SATs than in VATs[78], FAO-related mt-localized nu-PCGs were more abundant in SATs than in GOM and MAD but comparable to those in SAT-like RAD (Fig. 5e). These results suggested a higher mitochondrial respiratory capacity and a predominant role for SATs in metabolism, again supporting the SAT-like features of RAD (Supplementary Fig. 29d).

**Evolutionary dynamics of the transcriptome in animal models.** Pigs have emerged as an important biomedical model. To explore the evolutionary divergence of the transcriptome that contributes to tissue-specific biology of animal models, we performed comparative analyses of the transcriptomes of seven homologous tissues from pigs and eight representative vertebrate mammalian models including non-human primates (rhesus macaque), rodents (mouse, rat, and guinea pig), lagomorphs (rabbit), carnivores (dog and cat), artiodactylids (sheep), and a bird (chicken, an evolutionarily distant outgroup).

We first compared the transcription of 8428 single-copy orthologous PCGs as well as the alternative splicing patterns, or "percent-spliced in" (PSI) index values, of 15,172 orthologous exons shared by the nine mammals. We observed a tissue-dominated clustering pattern for PCG transcription (Fig. 6a) but species-dominated clustering patterns for alternative splicing[79,80] (Fig. 6b). Both unsupervised clustering and principal component analysis (PCA) recapitulated the distinct transcriptomic characteristics between transcription and alternative splicing (Fig. 6c–f). These

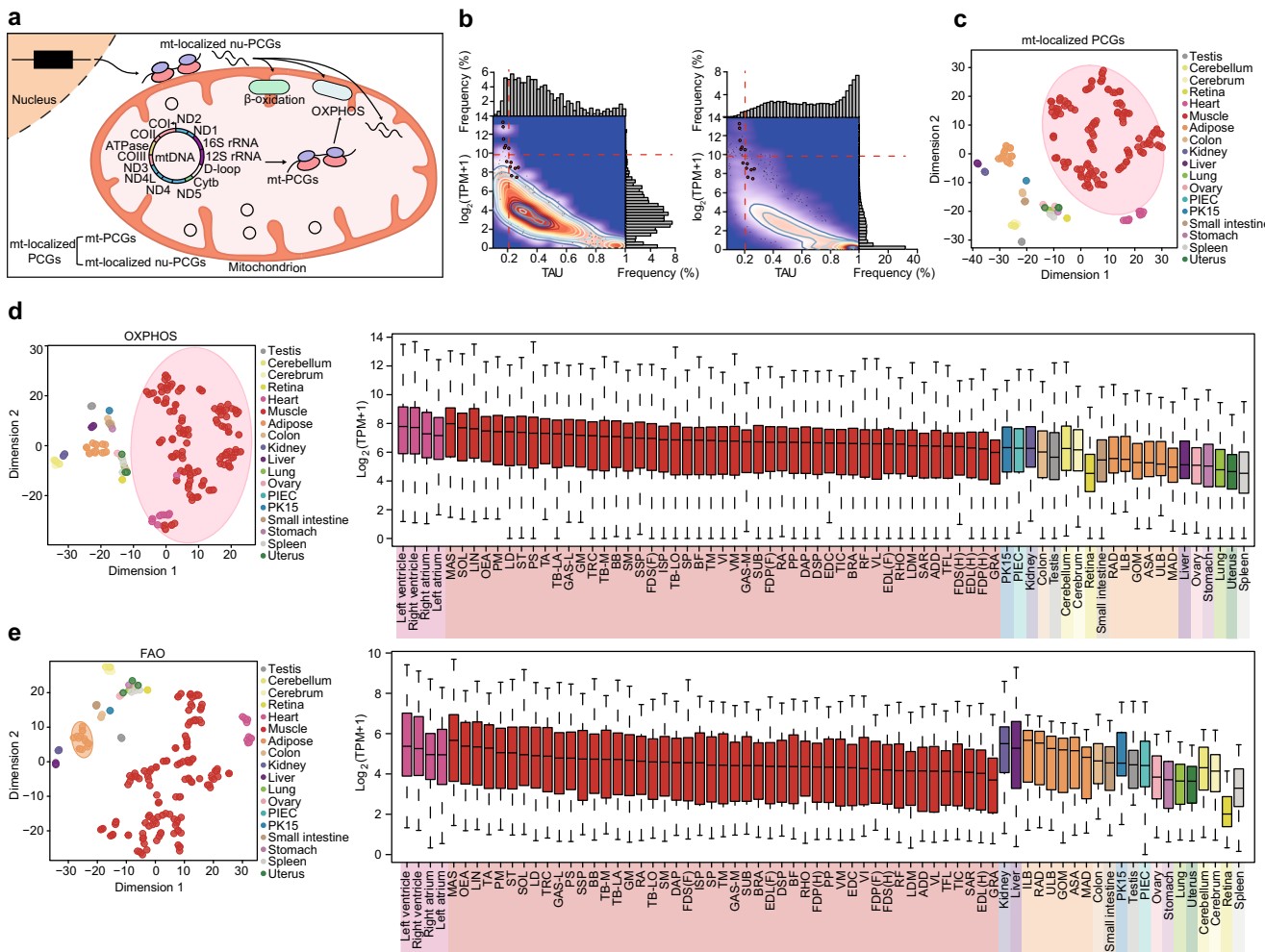

**Fig. 5 Transcriptional patterns of diverse mitochondrial pathways across tissues. a** Schematic representation of mitochondrial structure depicting the compendium of mitochondrial PCGs, including 13 PCGs encoded by mitochondrial DNA and 1043 post-translationally imported nuclear PCGs. mt-PCGs: mitochondrial protein coding genes; mt-localized nu-PCGs: mitochondria-localized nuclear PCGs. **b** Density/contour plots of the distribution of tissue specificity ($\tau$) and transcription level (TPM: transcripts per million) for PCGs located within (left panel) or outside of (right panel) the mitochondria. Data points in the plot indicate the 13 mt-PCGs, with the horizontal and vertical dashed lines indicating the average transcription level and tissue specificity of these mt-PCGs, respectively. **c** t-SNE plot depicting the transcriptional patterns of all mt-localized PCGs across 72 tissues. The transcription levels were log$_2$-transformed. **d** t-SNE plot showing clustering of the transcriptional patterns of mt-localized OXPHOS PCGs across tissues (left panel). Ellipse in the t-SNE plot indicates clustering of SMTs, constructed at a probability of 0.95. The boxplot indicates dynamic transcription levels (right panel, $n = 88$). In the boxplot, the internal line indicates the median, the box limits indicate the upper and lower quartiles and the whiskers extend to 1.5 IQR from the quartiles. **e** t-SNE plot showing clustering of the transcriptional patterns of the mt-localized FAO PCGs across tissues (left panel). Ellipse in the t-SNE plot indicates the AT cluster, constructed at a probability of 0.95. The boxplot indicates the dynamic transcription levels (right panel, $n = 36$). In the boxplot, the internal line indicates the median, the box limits indicate the upper and lower quartiles and the whiskers extend to 1.5 IQR from the quartiles. Source data for **b**–**e** are provided as a Source Data file.

results suggested that evolutionarily conserved transcriptional differences underlie tissue identity across mammals, whereas exon splicing may be more often affected by species-specific changes in *cis*-regulatory elements and/or *trans*-acting factors than transcription[79,80]. These results were also observed over large evolutionary time scales when using the transcription of 6433 single-copy orthologous PCGs and the alternative splicing patterns of 12,662 orthologous exons shared by nine mammals and chicken (Supplementary Figs. 34 and 35).

Notably, in the clustering patterns based on transcription, tissues of chicken formed a distinct cluster rather than grouping with their mammalian counterparts (Supplementary Fig. 35a), which was supported by the greater divergence in transcription between mammals and chicken (average Pearson's $r = 0.63$) than

between mammals (average Pearson's $r = 0.75$) for each tissue (Supplementary Fig. 34). Divergences in transcription were highly consistent with evolutionary divergence times between pairs of species for each tissue (average Spearman's $r = -0.68$), which indicated that divergence in transcription among these species started to surpass divergence between different tissues at approximately the same time proposed for the divergence of birds and mammals (~300 million years ago [MYA]). In contrast, among mammals, mouse, rat, and guinea pig (the most closely related rodent models, with divergence ~15.9 MYA for mouse and rat and ~70 MYA for mouse/rat and guinea pig) showed higher similarity in transcription in each tissue to each other (average Pearson's $r = 0.80$) than to other non-rodent mammals (average Pearson's $r = 0.74$, 0.75, and 0.75 for mouse, rat and

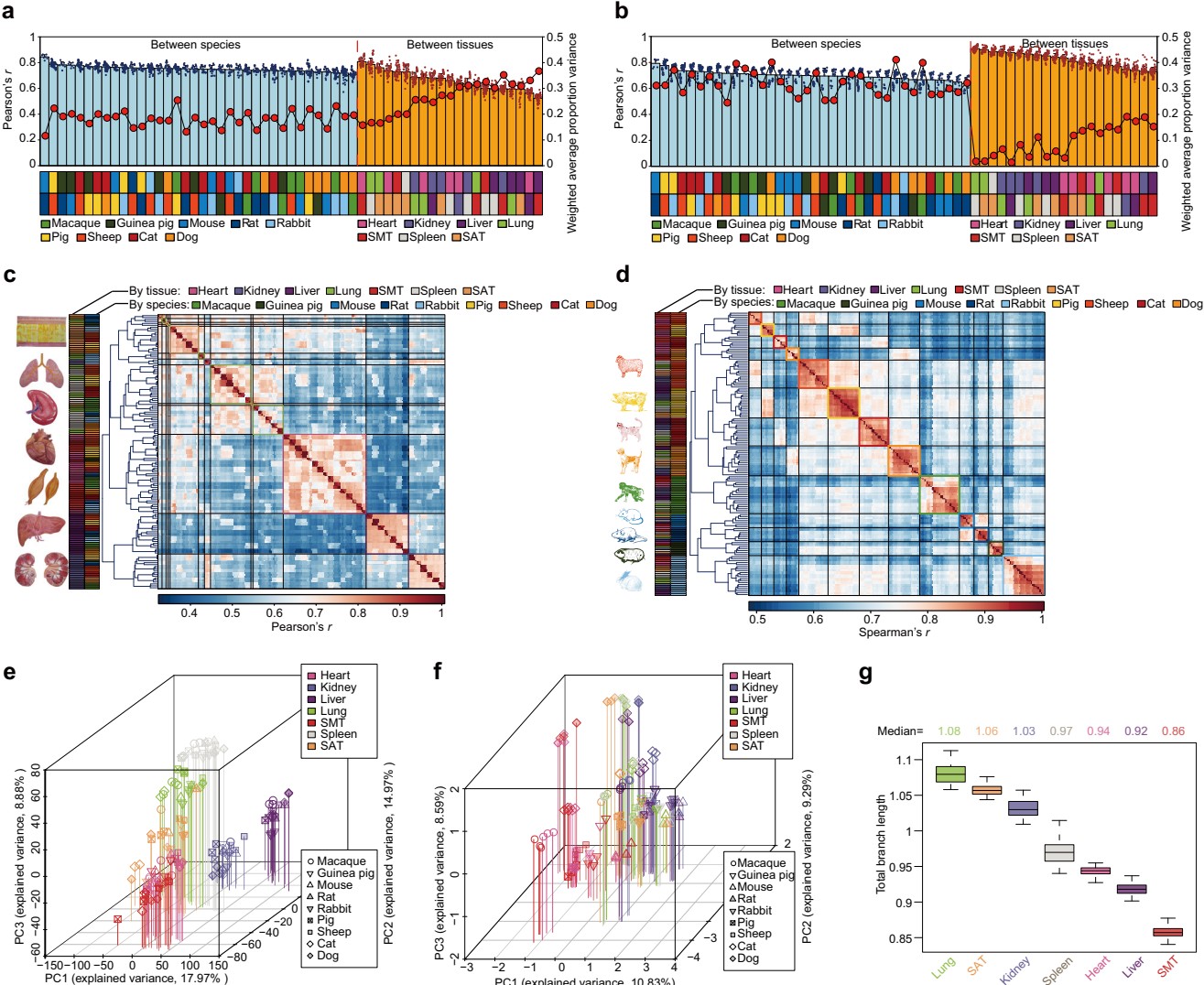

**Fig. 6 Global patterns of PCG transcription and alternative splicing across mammalian models. a, b** Comparison of variation between species (nine mammals) and tissues revealed by (**a**) transcription and (**b**) alternative splicing patterns. Bar plots represent pairwise Pearson's correlations between species (light blue bar) and between tissues (orange bar). The correlations were calculated based on the transcription levels of 8428 1-1 orthologs and 15,172 orthologous exons across nine mammals. Data are presented as mean values ± SD. Weighted average proportion variances (WAPVs) of PCG transcription and alternative splicing (reflected by PSI values) were determined using principal variance component analysis (PVCA) and are depicted as red dots connected by black lines. Boxes (bottom) indicate pairwise comparisons presented in a column according to the color assigned to each species or tissue. For PCG transcription, there were more profound transcriptional differences among tissues (Pearson's $r = 0.66$ and WAPV = 0.29) than among species (Pearson's $r = 0.75$, WAPV = 0.18). By contrast, for alternative splicing, the differences among species (Pearson's $r = 0.70$ and WAPV = 0.32) were greater than those among tissues (Pearson's $r = 0.82$ and WAPV = 0.10). **c, d** Hierarchical clustering analysis of samples using (**c**) PCG transcription and (**d**) alternative splicing (reflected by PSI values). Average linkage hierarchical clustering was based on distances between transcription levels of samples measured by Pearson's correlation. **e, f** Factorial map of the principal component analysis (PCA) of (**e**) PCG transcription levels and (**f**) alternative splicing. The proportion of variance explained by each principal component is indicated in parentheses. The vertical lines of different colors dropping from the plotted points to the x/y plane show the separation of points based on the first and second principal components. **g** Box plot depicting the total branch lengths of neighbor-joining trees of PCG expression (see Supplementary Fig. 37) constructed based on pairwise ($1−r$) distances (here, $r$ is Spearman's correlation coefficient) across nine mammals for each tissue. The distribution is based on random sampling ($n = 100$ replicates). In the boxplot, the internal line indicates the median, the box limits indicate the upper and lower quartiles and the whiskers extend to 1.5 IQR from the quartiles. Source data are provided as a Source Data file.

guinea pig, respectively) (Supplementary Fig. 36). Relatively high similarity of transcription was also observed between the more closely related dog and cat [carnivores] (diverged ~55 MYA) and between pig and sheep [artiodactylids] (diverged ~64 MYA) compared to their pairwise comparisons against other evolutionarily distant mammals (average Pearson's $r = 0.77$ between cat and dog vs. 0.74; Pearson's $r = 0.79$ between pig and sheep vs. 0.76) (Supplementary Fig. 36).

We next investigated the evolutionary divergence of transcription between mammalian tissues by measuring the total branch length of transcription-based trees for each tissue based on 8428 single-copy orthologous PCGs shared by the nine mammals (Fig. 6g, Supplementary Fig. 37) and found that lung tissue (total branch length = 1.08) and SAT (1.06) had longer total branch lengths than other tissues (kidney [1.03], spleen [0.97], heart [0.94], liver [0.92], and SMT [0.86]), implying a potentially higher

evolutionary rate than that of other tissues (Fig. 6g). These findings suggest that divergence in transcription is generally evolutionarily correlated but is slightly different for functionally distinct tissues.

Evolutionary pressures during speciation and adaptation are mainly expected to alter PCG transcription rather than changes in protein sequences[81]. In order to test how the evolution of PCG transcription is shaped by rapidly evolving regulatory elements (especially enhancers, key regulatory DNA elements that engage in physical contact with their target-gene promoters), we generated 13 in situ Hi-C maps of SAT for six mammals (a total of ~3.81 billion uniquely aligned contacts with a depth of ~293 M contacts per library) (Supplementary Tables 3–5). With six in situ Hi-C maps of SAT for pigs, we identified a repertoire of PCG promoter and enhancer interactions at 20 kb resolution for seven mammals (Supplementary Fig. 38, Supplementary Table 6). We found that the enhancer number for a PCG was moderately correlated across mammals ($r = 0.452$, Supplementary Fig. 39) and that PCGs associated with larger numbers of enhancers were more highly expressed in all species (average Spearman's $r = 0.848$, between enhancer number and transcription level for each species, Fig. 7a), suggesting that the majority of the enhancers identified by this analysis had a measurable additive effect on target-PCG transcription[82].

In comparison with their respective control PCGs matched by transcription level, we found that PCGs associated with multiple enhancers (≥5) or with only one or fewer enhancers (≤1) across species, respectively, showed significantly increased ($P = 2.86 \times 10^{-6}$, Wilcoxon signed-rank test) and decreased ($P = 0.035$) transcriptional conservation (determined by correlations with the transcription levels of 1557 and 1552 single-copy orthologous PCGs, respectively, between species) (Fig. 7b, c). Moreover, compared with PCGs with a low coefficient of variation (CV) for their transcription across species (i.e., evolutionarily stable), PCGs with a high CV (i.e., evolutionarily variable) exhibited only marginal differences in nucleotide sequence conservation estimated by phastCons values (0.11 vs. 0.13, $P = 0.063$) and phyloP values (0.13 vs. 0.21, $P = 0.188$) (Fig. 7d, Supplementary Fig. 40). This finding implied that the divergence of PCG transcription across species is independent of sequence conservation, that is, changes in PCG transcription across species could be largely driven by the sheer number of regulatory elements (enhancers) rather than by primary sequence characteristics. These observations also aligned well with the hypothesis that enhancer redundancy may contribute to regulatory innovation during evolution by allowing enhancer sequence variation to subtly alter, and thus stabilize, the transcription of functionally important target PCGs[83]. In contrast, PCGs with relatively few associated enhancers may have only limited buffering of their transcriptional regulation against genetic perturbations during speciation, which could thereby result in transcriptional changes/shifts across species[83,84].

## Discussion

The pig, particularly its miniature breeds, has recently emerged as a biomedical model for multiple complex diseases. The ability to generate genome-editing mutations in combination with somatic cell nuclear transfer procedures has yielded useful models for several human diseases. The pig is also a potential source for xenotransplantation due to its high anatomical, genetic, and physiological similarities to humans[85]. Our study greatly expands the annotation of transcripts in the reference pig genome and offers a comprehensive landscape of transcription across tissues/cell types with different physiologies. In particular, we focused on metabolically active SMTs and ATs, which are associated with obesity-related disorders, in addition to being economically important products.

Comparative transcriptomic analysis of homologous tissues between pigs and other widely used mammalian models can provide essential information for the use of this animal as a biomedical model, including its advantages and disadvantages. For instance, the PCGs with significant transcriptional shifts across species should be considered when selecting targets in animal models to extrapolate diseases or phenotypes. The evolutionary divergence of PCG transcription also provides a primary reference for the extent to which the biology of a given species can be extrapolated to another (Supplementary Fig. 41). Each animal model has unique strengths and weaknesses regarding the aspects of application.

In practice, large mammals have tissue sizes more comparable to those of humans, making them more viable as potential sources for xenotransplantation[86]. In particular, pigs are broadly available (especially because of their short generation times and large litter sizes), ethically more acceptable than canines and non-human primates, and generally more predictive of therapeutic treatment efficacy in humans than rodents[86,87]. Nonetheless, small mammals have been the preferred models for studies of human biology and diseases (e.g., laboratory rodent strains are extensively used for cancer, cardiovascular, and metabolic disease[88]). We identified a total of 5902 single-copy orthologous PCGs with significant species-specific transcriptional shifts in seven mammalian tissues (transcriptional changes after correcting for evolutionary nucleotide divergence between mammals) (Supplementary Fig. 41). These extensive PCG lists reflect biologically relevant differences in transcription in the indicated mammals (see Supplementary Data 7 for functional enrichment analysis), which should be considered when using these PCGs as targets in animal models. Literature searches provide insights into the potential functional implications of some of the significant transcriptional shifts. For example, CMYA5, which is associated with Duchenne muscular dystrophy[89], was specifically upregulated in SMTs of pigs (TPM = ~493.79) compared to those of other species (TPM = ~97.15), and nucleotide substitution in this PCG has been potentially associated in previous reports with carcass and meat quality traits in pigs[90]. ACE2, the SARS-CoV-2 receptor required for cell entry in humans[91] and a target of diabetes therapy, was specifically upregulated in the SATs of pigs (TPM = ~184.31) compared to those of other species (TPM = ~0.49); knockout of this PCG can worsen inflammation of ATs and exacerbate high-calorie diet-induced insulin resistance in mice[92].

Detailed analysis of the transcriptional landscape of pigs can thus provide a highly informative resource on transcript characteristics and offers system-wide insights into tissue-specific physiological activities. These results therefore serve as a prelude to advances in pig biology, as well as the use of pigs as model organisms for human biological and biomedical studies.

## Methods

**Animals and samples**. All research involving animals was conducted according to Regulations for the Administration of Affairs Concerning Experimental Animals (Ministry of Science and Technology, China, revised in March 2017), and approved by the animal ethical and welfare committee (AEWC) of Sichuan Agricultural University under permit No. DKY-B20171902. The animals were allowed access to feed and water ad libitum and were humanely killed as necessary to ameliorate suffering and were not fed the night before they were slaughtered. Collection and sequencing of human clinical samples were approved by the Ethics Committee of Sichuan Provincial People's Hospital (No. 2018–212), and informed consent was obtained before the study.

**Pig transcriptome reconstruction**. To comprehensively survey the pig transcriptome, a total of 194 samples from 70 tissues (1–3 biological replicates for each of 17 solid tissues, as well as 47 skeletal muscles and 6 adipose depots from different body sites) and two immortalized cell lines (PK15 and PIECs) were used in this study. Two cell lines, PK15 (catalogue no. KCB201002YJ) and PIEC

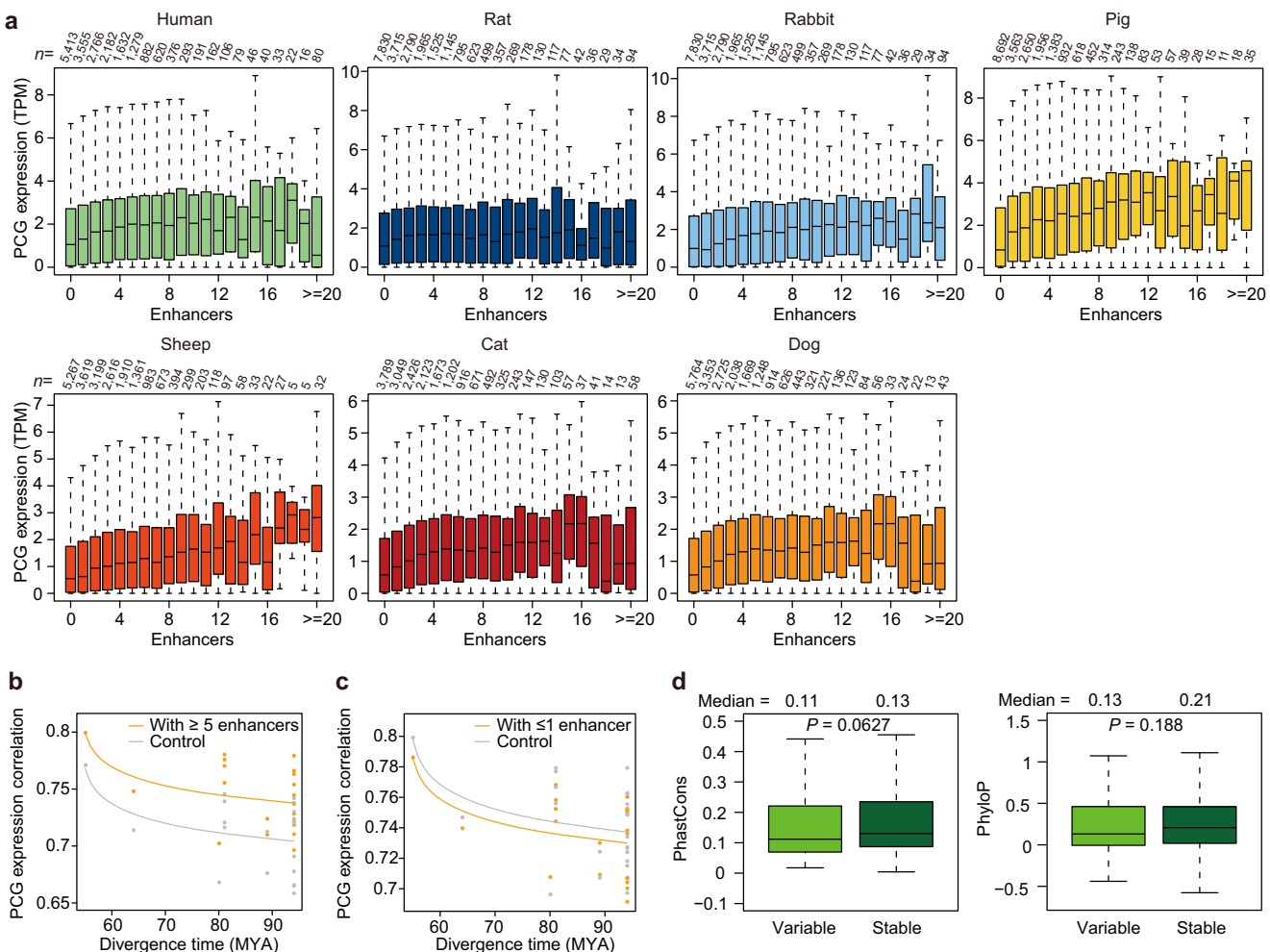

**Fig. 7 The number of enhancers drives the stability of PCG transcription across the mammalian phylogeny. a** Box plots showing distributions of PCG transcription associated with increasing numbers of enhancers in SAT in each mammalian species. Enhancers associated with a PCG have an additive effect on transcript levels. In the boxplot, the internal line indicates the median, the box limits indicate the upper and lower quartiles and the whiskers extend to 1.5 IQR from the quartiles. The numerical value above each bar indicates the number (*n*) of genes. **b, c** The number of associated/interacting enhancers contributes to the evolutionary stability (**b**) or viability (**c**) of PCG transcription. Pairwise Spearman's correlation coefficients of transcription levels between species were plotted against the evolutionary distance of PCGs associated with multiple (≥5) enhancers (1557 PCGs) and compared with control PCG sets that have only one or fewer enhancers (enhancers ≤1) but a matched transcription level (**b**). For contrast, plots using the same analysis but showing PCGs associated with only one or fewer enhancers (1552 PCGs) compared with control PCG sets with multiple enhancers (**c**). The number of associated/interacting enhancers corresponds to the median number across species. The lines correspond to linear regression trends (after log transformation of the time axis). MYA: million years ago. **d** PCGs with either stable or variable transcription are largely similar in nucleotide sequence conservation (phastCons value [left panel] and phyloP value [right panel]). Stable PCGs: *n* = 245; variable PCGs: *n* = 245. (two-sided Wilcoxon rank sum test, Bonferroni-corrected *P*). In the boxplot, the internal line indicates the median, the box limits indicate the upper and lower quartiles and the whiskers extend to 1.5 IQR from the quartiles. Source data are provided as a Source Data file.

(catalogue no. GNO15), were obtained from the China Infrastructure of Cell Line Resources and Stem Cell Bank of the Chinese Academy of Sciences, respectively.

We constructed rRNA-depleted and random priming RNA-seq libraries and sequenced them on the Illumina HiSeq X Ten platform to produce an average of ~49 million 150-bp paired-end raw reads and ~48 million high-quality reads for each library. Sequenced reads were aligned to the pig reference genome (Sscrofa 11.1, GCA_000003025.6) by the STAR alignment tool (version 2.5.3a)[93], with on average ~96% (~85.76 million) of aligned reads for each individual library.

For unbiased representative construction of the pig transcriptome, 33 tissues/cell lines (87 samples) comprising the core atlas dataset were chosen for de novo transcriptome assembly. The aligned reads of these samples were assembled with Cufflinks (version 2.1.1)[94]. We employed a previously reported computational method to filter out library-specific background noise and predict the most likely isoforms from the assemblies of transcript fragments (transfrags). We obtained high-quality assemblies containing reliable transcripts which were further submitted to construct transcriptome maps by TACO[95] (a meta-assembly method with a robust solution for leveraging the vast RNA-seq data landscape for transcript structure prediction).

To facilitate further analysis of the pig transcriptome, we estimated the coding potential of putative non-coding transcripts that were not annotated as PCGs in the

pig reference genome using a stringent filtering pipeline and classified them into long noncoding RNAs (lncRNAs) and transcripts of unknown coding potential (TUCPs). For circRNA prediction, we retrieved RNA-seq reads that were mapped to back-splicing junction sites using CIRCExplorer2 (version 2.3.2)[96], with putative circRNAs required to have more than two independent junction-spanning reads. Correspondingly, we also performed small RNA sequencing. We used pig miRNAs annotated in the miRbase[19] database, as well as putative miRNAs annotated using other mammalian and avian miRNA sequences as references that were not present in the current pig genome.

Gene-level transcript abundance was estimated as transcripts per million (TPM) using the high-speed transcript quantification tool Kallisto (version 0.43.0)[97]. For PCGs, TUCPs, and lncRNAs, we considered a gene as detected/transcribed if it had an expression value greater than 0.1 TPM in at least one sample. For miRNAs and circRNAs, we used a cut-off of 1 TPM and 0.05 TPM in at least one sample, respectively.

**Reconstruction of 3D genome structures**. To visualize the location dependence of transcription within the nuclear space, we performed an in situ Hi-C experiment for a

subcutaneous AT (ULB, upper layer of backfat) of six individuals. Hi-C data were processed using a custom pipeline implemented in Juicer software (version 1.8.9)[98].

We reconstructed three-dimensional (3D) genome structures with both intra- and interchromosomal interactions at a 100 kb resolution using miniMDS[99] (version 2018-09-27). We also visualized the 3D genome using 3D modeling.

To investigate the correlation between transcription and 3D genome folding, we generated the landscape of chromatin organization including compartment A/B[100] (at 20 kb resolution and 100 kb resolution for visualization) and topologically associating domains (TADs)[101] (at 20 kb resolution) for the pig transcriptome.

Genomic properties, including GC content and gene density, were estimated using 100 kb windows and further integrated with the 3D genome structures.

See Supplementary Methods for more details.

**Gene transcriptional profiling across tissues**. We calculated the tissue specificity of gene abundance reflected by the tau score ($\tau$)[102] (ranging from 0 to 1, with 1 for highly tissue-specific genes and 0 for ubiquitously transcribed genes) for each gene with scaled TPM values. For each tissue, we averaged all replicates and then calculated $\tau$ to account for unequal numbers of replicates among tissues. We used $\tau \geq 0.75$ as the cut-off for tissue-specific genes.

We calculated the abundance distribution (i.e., transcriptome complexity) of distinct transcripts across tissues, reflected as the fraction of total RNAs contributed by the most highly expressed genes.

Differential gene expression analysis was performed using edgeR (version 3.22.5)[103], with a false discovery rate (FDR) $\leq 0.05$ and $\log_2$(fold change) $\geq 1$ as cut-offs for statistical significance.

**Data collection for the functional gene categories**. To further characterize the specialized functions of different tissues in this study, we collected multiple a priori functional candidate PCGs and examined their expression patterns. PCGs involved in the core functions of SMTs (i.e., 'HOX genes', 'homeobox family genes', 'myo-kines', and 'NMJs'), ATs ('inflammatory response', 'extracellular matrix', 'HOX genes', 'homeobox family genes', and 'glucose and lipid metabolism genes') were retrieved from public databases (Kyoto Encyclopedia of Genes and Genomes (KEGG) and Gene Ontology) and/or collected from the literature.

The entries of PCGs that encode mitochondrially localized proteins were extracted from the MitoCarta (version 2.0) database[75]. The gene inventories of two central metabolic pathways, OXPHOS and FAO, were retrieved from the Gene Ontology database.

**Spatial transcriptomics (ST) of SMTs and myofiber composition estimation**. We performed ST using the representative psoas major muscle (PM) in two replicates. Tissue sections on Visium Spatial slides were permeabilized according to the protocol provided by 10X Genomics. ST cDNA libraries were sequenced on the Illumina NovaSeq 6000 platform using paired-end sequencing, producing a mean depth of 50 million paired-end reads, which resulted in an average library saturation above 90%. After normalization, we performed dimensionality reduction and visualized spot clusters in a reduced 2D space using UMAP[104]. We determined the type I and II muscle fiber clusters and then discriminated types IIA and IIB within the type II cluster. We visualized the spatial distribution of each cluster using the SpatialPlot function and compared it with the ATPase staining myofiber (type I) pattern. With ST data, we estimated the myofiber proportion using bulk RNA-seq data for different types of SMTs derived from distinct body parts using CIBERSORTx (in silico deconvolution, https://cibersortx.stanford.edu/)[57].

See Supplementary Methods for more details.

**Estimation of relative cell-type proportions in adipose tissues**. We applied CIBERSORTx[57] and the adipose tissue signature matrix to estimate the relative cell-type proportions in adipose tissues. The CIBERSORTx adipose tissue signature matrix was obtained from a previously published study in humans[65], which included purified cells that are known to be present in adipose tissue, including adipocytes, macrophages, CD4$^+$ T cells, and microvascular endothelial cells (MVECs). A total of 571 genes in this signature matrix with pig orthologs were used for subsequent analysis.

**Comparative transcriptomic analysis across species**. We also sequenced 142 rRNA-depleted RNA-seq libraries of seven homologous organs/tissues (adipose, heart, kidney, liver, lung, skeletal muscle, and spleen) from eight major mammalian models (macaque, mouse, rat, guinea pig, rabbit, cat, dog, and sheep) and chicken using similar library construction procedures and the same sequencing platform used for pig samples. Sequenced reads were aligned to corresponding reference genomes (macaque: Mmul_8.0.1, rabbit: OryCun2.0, mouse: GRCm38.p5, rat: Rnor_6.0, guinea pig: Cavpor3.0, sheep: Oar_v3.1, dog: CanFam3.1, cat: Felis_catus_6.2, and chicken: Gallus_gallus-5.0) by the STAR alignment tool (version 2.5.3a). Quantification of PCG expression in each species was performed similarly to that used in the pig analyses.

We performed comparative analyses of gene transcription and alternative splicing across pigs and 9 other species based on single-copy orthologous PCGs and orthologous exons. Single-copy orthologous PCG families were identified following the protocol recommended by Ensembl (http://asia.ensembl.org/info/genome/compara/homology_method.html). Cross-species comparative analyses of alternative splicing were performed based on the "percent-spliced in" (PSI) values of orthologous exons.

We performed selection analysis on transcriptional abundance for individual PCGs based on Ornstein-Uhlenbeck (OU) and Brownian motion (BM) models, as in previous reports[105].

Additional details for the process are provided in the Supplementary Methods.

**Transcription divergence and PEIs across species**. We obtained 13 Hi-C datasets of SAT from six mammals, including human, rat, rabbit, cat, dog, and sheep, following the experimental and analytical procedures described in 'Hi-C experiment and data analysis' in the Supplementary Methods.

PSYCHIC[106] (version 2018-01-05) was applied to identify overrepresented promoter-enhancer interactions (PEIs) at a resolution of 20 kb using interaction intensity normalized according to the background model.

Evolutionary divergence of transcription was measured by Spearman's correlation coefficients for the transcription levels of orthologous PCGs between pairs of species.

When estimating the relative divergence of PCGs with different numbers of interacting enhancers, confounding effects due to differences in transcription level distributions were controlled for by matching PCGs one-to-one to control PCGs with similar expression using the MatchIt library in R.

**Reporting summary**. Further information on research design is available in the Nature Research Reporting Summary linked to this article.

## Data availability
Raw and processed RNA, and miRNA sequencing data of pigs have been deposited in the NCBI Gene Expression Omnibus (GEO) under accession codes "GSE162145" and "GSE162147", respectively. Spatial transcriptomic data have been deposited in GEO under accession code "GSE161882". The raw and processed RNA sequencing data of nine other species for comparative transcriptomic analysis have been deposited in GEO under accession code "GSE162142". RNA-sequencing and Hi-C data of non-human species for analysis of gene transcription divergence and PEIs across species have been deposited in GEO under accession codes "GSE162146" and "GSE162140", respectively. Human RNA sequencing and Hi-C data have been deposited in GEO under accession codes "GSE162143" and "GSE162139", respectively. The nuclear PCG entries with mitochondrial localization were downloaded from MitoCarta2.0 database (http://www.broadinstitute.org/pubs/MitoCarta). All other data supporting the findings of this study are available within the article and its Supplementary Information files or from the corresponding author upon reasonable request. A reporting summary for this Article is available as a Supplementary Information file. Source data are provided with this paper.

## Code availability
Each use of software programs has been clearly indicated and information on the options that were used is provided in the Methods and the Supplementary methods section. All software, codes, and scripts used for data processing and analyses are available on GitHub through the following link, https://github.com/QianZiTang/, or at https://doi.org/10.5281/zenodo.4724411.

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

## Acknowledgements

This work was supported by grants from the National Key R & D Program of China (2018YFD0500403, 2020YFA0509500, and 2018YFD0501204), the National Natural Science Foundation of China (U19A2036, 31872335, 31772576, and 31802044), the Sichuan Science and Technology Program (2021YFYZ0009 and 2021YFYZ0030), and the Earmarked Fund for the China Agriculture Research System (CARS-35-01A).

## Author contributions

M.L., Q.T., and L.J. led the experiments and designed the analytical strategy. L.J., Y.W., M.H., L.S., K.L., J.M., X.W., Z.C., G.L., ZL.C., and Y.J. performed animal work and prepared biological samples. M.H., L.J., Z.C., and X.Z. constructed the sequencing library and performed sequencing. Q.T., M.L., L.J., Z.C., and B.Z. designed the bioinformatics analysis process. Q.T., Z.C., L.G., G.T., F.L., and B.Z. performed the transcriptome assembly. S.H., Z.C., X.Z., and D.L. performed the transcripts annotation and expression quantification analysis. L.J., Q.T., M.H., Y.L., and L.Z. performed 3D genome structures reconstruction. L.J., S.H., and Z.C. performed the gene transcriptional profiling. L.J., K.L., Q.T., and B.Z. performed spatial transcriptomics experiment and analysis. Q.T., S.H., L.J., and M.L. performed comparative transcriptomic analysis. L.J., Q.T., X.Z., and Y.G. performed joint analysis of transcription divergence and PEIs. L.J., Q.T., S.H., Z.C., X.Z., and M.L. wrote the paper. L.J., M.L., X.Z., V.N.G, J.Y., Z.H., and X.L. revised the paper.

## Competing interests

The authors declare no competing interests.
