## [Peer Review File · Nature Communications]

Reviewers' Comments:

Reviewer #1:

Remarks to the Author:

This manuscript describes the collection and analysis of a tremendous amount of functional genomic data in pig. The amount of data that was collected and reported in this study is simply tremendous - several hundred RNA-seq datasets analyzing 70 tissues, a spatial transcriptomics dataset, and in situ Hi-C data from several tissues, which were collectively sequenced to a depth of many billions of reads. The data collection was primarily centered around muscle and adipose tissues, which has a high relevance for agricultural industries. Finally, the authors also collected new data in a cohort of mammals and conducted an interesting comparative analysis. These datasets themselves appear to be of high quality (based on the QC metrics reported in the paper) and are likely to be highly important resources for pig genetics. In addition, the analyses are also of high quality, convincing, and very well described in the manuscript. I am in favor of publication of this paper.

Comments:

1. The Visium spatial transcriptomics (ST) system used by the authors has spots that are quite large, which in many tissues provides data spanning across multiple spatially clustered cell types. The authors use the ST data as the basis for their analysis of differential gene expression and muscle fiber deconvolution analyses, which use methods that in many respects require single cells. Are muscle fibers large enough that ST data provides single-cell resolution at the current visium scale? If so, I think it would be helpful to state this explicitly in the Results section when introducing this analysis. If not, I think the methods used by the authors are likely to largely be appropriate, however the assumptions underlying them should be clearly stated.
2. The authors use human data to provide a reference for cell type deconvolution in pig. While the authors perform this analysis carefully, using marker genes, the expression of these markers may very well have nevertheless changed in the orthologous porcine cell type. One strategy to make sure this is not a major issue is to analyze the robustness to differences in which markers are used in the deconvolution analysis. This analysis could simply involve picking a random subset of markers X% of the time (similar to a bootstrap analysis), and use this to provide a confidence range of cell type proportions. This type of analysis would strengthen the authors' points by demonstrating that major results are not driven by a single marker (or a small collection) that has changed expression in pig compared with human.
3. The authors conduct an interesting comparative analysis of enhancers in five species. The authors used data from several new in situ Hi-C libraries to estimate enhancer-promoter interactions. It's not clear to me (from either the results or methods sections) where the location of enhancers were obtained from. Please clarify where enhancers were obtained from.

Reviewer #2:

Remarks to the Author:

Major comments:

The manuscript by Jin et al. entitled "A Pig BodyMap Transcriptome Reveals Diverse Tissue Physiologies and Evolutionary Dynamics of Transcription" is data and information rich. After the BodyMaps of the human, mouse, and rat, the pig BodyMap reported here debuted large numbers of TUCP and CircRNA, together with large numbers of new miRNAs and lncRNAs. The study should serve as an important reference for the scientific communities in both agriculture and biomedical research. In this regard, it has a wide readership and should be published in a high impact journal. The manuscript, however, is not written to the standards of Nature communication, albeit mostly understandably and logically presented. There are numerous grammatical errors, redundancies, long sentences coupled with inappropriate and misplaced prepositions. The authors should fully spell out all tissues names. The acronyms are unusual and together with grammatically problems, the text is very difficult to read. As bar plots or heatmaps, the GO plots

contain only p-values. This website, <https://bioc.ism.ac.jp/packages/3.7/bioc/vignettes/enrichplot/inst/doc/enrichplot.html#dot-plot>, employs enrichGO plots. This type of plot includes both the p-values and the numbers of genes for each GO category and is therefore more informative. The authors should use this type of plot in at least some figures in this manuscript. Please commit that your analysis pipeline code are published on Github.

A main question is: the study collected multiple tissues of all three germ layers. Are there any commonalities in transcription among tissues from the same germ layer? And similarly, what are the main differences among different germ layers?

Minor comments:

Fig. 1: the term "transcribed transcript" is very confusing. Are there un-transcribed transcripts? In Table 1, "detected transcripts" may be a better term. For the rest of the text, simply "transcript" is sufficient.

Fig 2. If the study includes 20 chromosomes for the pig (from Sup Fig. 1), why are X and Y not in this figure?

L154-155: Are these TPM values regarded as high, medium, low? Please add in Table 1 the ranges of TPMs for PCGs and the other types of transcripts.

L162-166: this statement is more than 4 lines' long. It is also written in a very convoluted and confusing fashion. Please revise.

L170: what is the "x" for?

L192: what are these "individuals"? They are all from one pig.

L209-210: statement not finished.

L244: axis of ?

L274: Only "hundreds of SMTs are controlled by the nervous system"? My understanding is that all of them are. Please check this.

L304-309: another long and confusing sentence that needs re-written.

L315-319: again long and poorly written sentence.

L328: the term "myofiber-specific associated" is very confusing. If the transcripts are myofiber-specific, why add "associated"?

L551: instead of calling the chicken a "group", "an evolutionally distant species" may be more appropriate.

L625-626: Please use short phrases to explain "phastCons values" and "phyloP values". It will also help the reader if you explain why they imply that the divergence of PCG transcription across species is independent of sequence conservation.

L641: somatic cell nuclear transfer is the formal scientific term for cloning.

L680, 687: "in pig" is not correct. When referring to a species, it should be either "in pigs" or "in the pig".

L881-883: For the three breeds of Chinese pigs used, please provide references that describe their physiological/production characteristics and genetic backgrounds.

Sup Fig. 2. No figure legend.

Sup Fig. 4a. Red and yellow could not be distinguished, change the color of one of them to a different color, black, for example.

Sup Fig. 5a. Please explain what the different colored lines are. If they denote genomic DNA, why do they have different colors? Also, "intergenic LncRNA" is used in Fig.5a, while in Fig. 5b. it is changed to LincRNA. If these are the same, please be consistent. Fig. 5b. There should be a space separating a bracket from the preceding text. This is a recurring issue in the entire supplementary file. Please change them all.

Sup Fig. 6a and b. The color keys in 6a are not the same as those in the diagram. Both a and b should be made three dimensional with multiple X-axes so the data can be better discerned. Fig. 6e: please explain the color scale, density of ? Is this a log scale?

Sup Fig. 7a. % of Relative abundance of cirRNA in ? in total? 7c: please explain Q1-Q4 here. What do these TPM values imply? If TPM>0.1 is regarded as transcribed, are these still transcribed?

Sup Fig. 12b-e: Please add the color keys.

Sup Fig. 16b-f: please explain the color scales and the dots in the plots. The density scales on the X- and Y-axes are in decimal and percentage, respectively. Shouldn't they be in one consistent format? L319: "between SMTs (Supplementary Fig. 16 b-f)". These figures do not show which pairs of SMTs were compared, neither in the plots nor in the legend.

Sup Fig. 22. Both panels of the figure are very small and it is not possible to see the GO terms in b. What do the color shades indicate on the X- and Y-axes in b?

Sup Fig. 24: please explain the Z-score.

Sup Fig. 28a: Please provide the color keys for the samples (lanes).

Sup Fig. 33. Please add a statement summarizing the data presented.

Reviewer #3:

Remarks to the Author:

The authors present a detailed analysis of an extensive pig transcriptome dataset. The manuscript is largely descriptive but nevertheless is comprehensive and very thorough. These data and the associated analyses represent a very valuable resource for researchers working on pigs both as biomedical models and as an important agricultural animal.

The major issue that needs to be addressed prior to acceptance is the availability of the data. The authors cite accession numbers for NCBI (PRJNA637678 and PRJNA655361) and National Genomics Data Center of China (<http://bigd.big.ac.cn/gsa-human>) (PRJCA003737) but searches of these data repositories with these accession numbers return no results.

The following would be more appropriate references in relation to the FAANG project (line 68): Andersson L, Archibald AL, Bottema CD, Brauning R, Burgess SC, Burt DW, Casas E, Cheng HH, Clarke L, Coudrey C, Dalrymple BP, Elsik CG, Foissac S, Giuffra E, Groenen MA, Hayes BJ, Huang LS, Khatib H, Kijas JW, Kim H, Lunney JK, McCarthy FM, McEwan JC, Moore S, Nanduri B, Notredame C, Palti Y, Plastow GS, Reecy JM, Rohrer GA, Sarropoulou E, Schmidt CJ, Silverstein J, Tellam RL, Tixier-Boichard M, Tosser-Klopp G, Tuggle CK, Vilkki J, White SN, Zhao S, Zhou H; FAANG Consortium. Coordinated international action to accelerate genome-to-phenome with FAANG, the Functional Annotation of Animal Genomes project. *Genome Biol.* 2015 Mar 25;16(1):57. doi: 10.1186/s13059-015-0622-4. PMID: 25854118; PMCID: PMC4373242.

Foissac S, Djebali S, Munyard K, Vialaneix N, Rau A, Muret K, Esquerré D, Zytnicki M, Derrien T,

Bardou P, Blanc F, Cabau C, Crisci E, Dhorne-Pollet S, Drouet F, Faraut T, Gonzalez I, Goubil A, Lacroix-Lamandé S, Laurent F, Marthey S, Marti-Marimon M, Momal-Leisenring R, Mompert F, Quéré P, Robelin D, Cristobal MS, Tosser-Klopp G, Vincent-Naulleau S, Fabre S, Pinard-Van der Laan MH, Klopp C, Tixier-Boichard M, Acloque H, Lagarrigue S, Giuffra E. Multi-species annotation of transcriptome and chromatin structure in domesticated animals. *BMC Biol.* 2019 Dec 30;17(1):108. doi: 10.1186/s12915-019-0726-5. PMID: 31884969; PMCID: PMC6936065.

Lines 105-106:

It appears that the comparison with the Ensembl annotation of the Sscrofa11.1 reference genome that is highlighted in lines 105-106 concerns Ensembl Release 90, August 2017 that included only 361 annotated lncRNA genes. The paper by Warr et al. 2020 that describes the Sscrofa11.1 reference genome cited Ensembl Release 98, September 2018 that included annotation for 6,798 lncRNA genes.

Lines 112-113

Similarly, it appears that the comparison with the Ensembl annotation of the Sscrofa11.1 reference genome that is highlighted in lines 112-113 concerns Ensembl Release 90, August 2017 that included 22,452 protein coding genes. The paper by Warr et al. 2020 that describes the Sscrofa11.1 reference genome cited Ensembl Release 98, September 2018 that included annotation for 21,301 protein coding genes.

Regardless there is an inconsistency between the text in line 112 that refers to 22,452 PGC and Supplementary Figure 1 that cites 22,342 PGC

The authors included two pigs cell lines in their study kidney epithelial cells [PK15] and iliac endothelial cells [PIECs] but do not discuss to what extent these cells provide a good representation of the original source tissues, i.e. kidney and ileum.

It would be useful to state whether the cDNA / RNA-Seq libraries were oligodT or random primed.

The following papers are also relevant to the characterization of the pig transcriptome:

Li Y, Fang C, Fu Y, Hu A, Li C, Zou C, Li X, Zhao S, Zhang C, Li C. A survey of transcriptome complexity in *Sus scrofa* using single-molecule long-read sequencing. *DNA Res.* 2018 Aug 1;25(4):421-437. doi: 10.1093/dnares/dsy014. PMID: 29850846; PMCID: PMC6105124.

Beiki H, Liu H, Huang J, Manchanda N, Nonneman D, Smith TPL, Reecy JM, Tuggle CK. Improved annotation of the domestic pig genome through integration of Iso-Seq and RNA-seq data. *BMC Genomics.* 2019 May 7;20(1):344. doi: 10.1186/s12864-019-5709-y. PMID: 31064321; PMCID: PMC6505119.

Summers KM, Bush SJ, Wu C, Su AI, Muriuki C, Clark EL, Finlayson HA, Eory L, Waddell LA, Talbot R, Archibald AL, Hume DA. Functional Annotation of the Transcriptome of the Pig, *Sus scrofa*, Based Upon Network Analysis of an RNAseq Transcriptional Atlas. *Front Genet.* 2020 Feb 14;10:1355. doi: 10.3389/fgene.2019.01355. PMID: 32117413; PMCID: PMC7034361.

Reviewer #1**Comment 1-1:**

This manuscript describes the collection and analysis of a tremendous amount of functional genomic data in pig. The amount of data that was collected and reported in this study is simply tremendous - several hundred RNA-seq datasets analyzing 70 tissues, a spatial transcriptomics dataset, and in situ Hi-C data from several tissues, which were collectively sequenced to a depth of many billions of reads. The data collection was primarily centered around muscle and adipose tissues, which has a high relevance for agricultural industries.

Finally, the authors also collected new data in a cohort of mammals and conducted an interesting comparative analysis. These datasets themselves appear to be of high quality (based on the QC metrics reported in the paper) and are likely to be highly important resources for pig genetics.

In addition, the analyses are also of high quality, convincing, and very well described in the manuscript. I am in favor of publication of this paper.

Response 1-1:

Many thanks for your constructive remarks in support of our study.

Comment 1-2:

1. The Visium spatial transcriptomics (ST) system used by the authors has spots that are quite large, which in many tissues provides data spanning across multiple spatially clustered cell types. The authors use the ST data as the basis for their analysis of differential gene expression and muscle fiber deconvolution analyses, which use methods that in many respects require single cells.

Are muscle fibers large enough that ST data provides single-cell resolution at the current visium scale? If so, I think it would be helpful to state this explicitly in the Results section when introducing this analysis. If not, I think the methods used by the authors are likely to largely be appropriate, however the assumptions underlying them should be clearly stated.

Response 1-2:

We highly appreciated this thoughtful comment.

We totally agree with the reviewer's concerns about whether spatial resolution is fine enough to resolve individual myofibers using current spatial transcriptomics (ST) technology.

Initially, we investigated the cross-sectional area of myofiber in skeletal muscle of pig. As shown in **Figure R1**, the median cross-sectional area of adult pig muscle fibers is $\sim 6951 \mu\text{m}^2$, which is ~ 2.93 -fold larger than the spot area of the current visium platform ($\sim 2,374 \mu\text{m}^2$), with $\sim 89.6\%$ myofibers exceeding the spot area, but smaller than the spot area of first-generation ST technology ($\sim 7848 \mu\text{m}^2$)¹. This result showed that pig muscle fibers are large enough that ST data could provide single-cell resolution at the current visium scale in our study.

Figure R1. Distribution of cross-sectional area of myofibers in pig skeletal muscle.

Cross-section of myofibers (144 myofibers) were from pig *psoas* major muscle, visualized for quantification by hematoxylin-and-eosin staining. The red vertical dashed lines indicate the median cross-sectional area of myofiber. The yellow and black vertical lines indicate the spot area of the current visium platform and the first-generation ST approach, respectively. The spot area was calculated as $\pi \times r^2$, where r is the spot radius.

In this section of the analysis, we aimed to identify myofiber-specific markers/signatures by selecting the most representative spots for three types of myofiber clusters.

Enlightened by the reviewer's suggestion, we developed a method for scoring local homogeneity (LH) for each spot (reflected by the number of each type of

six spots surrounding a given central spot) to assess the homogeneity of spots (*i.e.*, to ensure that the gene expression profile for a given spot was obtained from a homogeneous set of myofibers), as myofiber distribution exhibited a non-random pattern, with the same type of myofibers generally gathered together spatially.

Using this method, we updated our results regarding the selection of the most representative spots for the three types of myofiber clusters (**newly added Supplementary Fig. 23 and fig 24**). We have provided a detailed description of this process in the **Supplementary Methods**.

Newly added Supplementary Fig. 23. Assessment of homogeneity of

myofibers/spots based on local diversity of myofiber types.

a, Schematic diagram for scoring local homogeneity (LH) of type I and type II myofibers, respectively. The LH value of the center spot is calculated based on the homogeneity relationship of the 6 spots surrounding it. For example, for type I myofibers, if all 6 adjacent spots homogeneously belong to type I, the LH value of the center type I spot is calculated as 6. Therefore, all the spots were classified into six groups from 6 to 1. Spots with LH 0, indicating that none of the surrounding spots belong to the same type, were excluded.

b, Comparison of the spatial positions of different LH-grouped spots. Spots of each LH group (yellow or green dots) are overlaid to the ATPase-stained image (background), where dark and light areas indicate type I and type II myofibers, respectively. Two or three representative fields of view are shown for each group. The results were generated using two replicates. The results of type I muscle fiber image analysis show that the spot groups with higher LH values (6, 5) exactly overlap with dark stained areas where the type I muscle fibers were located. As the LH score decreases, the spots gradually shifted outward to the light-colored area (type II spot areas).

c, Distribution of relative expression levels of myofiber marker genes (*i.e.*, *MYH7* for type I, *MYH2* for type IIA, and *MYH4* for type IIB) across different LH groups classified using the LH scoring index shown in (a), for each of three types of myofiber. The expression levels (UMI counts) were normalized to the minimum expression of six LH groups.

Newly added Supplementary Fig. 24. Marker gene expression in representative spots for three types of myofiber clusters. Bar plots show higher expression level of marker genes in representative spots (top 200 spots with highest LH value) compared to 200 spots with lowest LH values. Error bars represent standard deviation (SD). Mann Whitney test was used to determine significance. *, $P < 0.05$; **, $P < 0.01$. The results were derived from two replicates.

Added Supplementary Method

“Selection of representative spots for three types of myofiber clusters

Myofiber distribution exhibited a non-random pattern, that is, the same type of

myofibers generally gathered together spatially. In light of this phenomenon, we developed a local homogeneity (LH) score for each spot (reflected by the number of each type of six spots surrounding a given central spot) to assess the homogeneity of spots (*i.e.*, to ensure that the gene expression profile for a given spot was obtained from a homogeneous set of myofibers) **(Supplementary Fig. 23a)**.

Using type I muscle fibers as an example, if the 6 adjacent spots all belong to type I, then the LH value of the center spot is calculated as 6, indicating a high probability that the central spot is also a type I myofiber due to the non-random pattern of homogeneous myofiber distribution.

According to this principle, we divided the three types of myofiber spots into 6 groups from LH 6 to 1. To verify the accuracy of the LH scoring, we compared the spatial position results of different LH groups with the results of ATPase-stained images **(Supplementary Fig. 23b)**. Spots with different LH scores were then mapped to their corresponding positions on the spatial slide using the `SpatialDimPlot()` function in Seurat. Then, the spots of each LH group were projected over an ATPase-stained image of the same region of the sample using an image mask. Moreover, we also investigated changes in the expression of the myofiber marker genes (*MYH7*, *MYH2*, and *MYH4*) among the different LH groups **(Supplementary Fig. 23c)**.

To accurately obtain the specific expression signatures of the three muscle fiber types, we extracted the top 200 spots with the highest LH scores for each type of myofiber, and used these as the most representative spots (*i.e.*, spots with highest possibility of accurately representing each type of myofiber) **(Supplementary Fig. 24)**.

Accordingly, we updated the results section with estimations of the proportions of each myofiber type in bulk RNA-seq data using deconvolution analyses, as shown in **revised Fig. 4e**, as well as the results in **revised Supplementary Fig. 25**. Briefly, the updated estimates of each myofiber proportion showed a high correlation to the results in the initial manuscript (Pearson's $r = 0.87$, $P = 4.38 \times 10^{-15}$ for type I; $r = 0.59$, $P = 1.49 \times 10^{-5}$ for type

IIA; $r = 0.96$, $P < 10^{-16}$ for type IIB). These updated results shown in **revised Supplementary Fig. 25** still support the conclusion that the proportion of type I myofibers estimated using the assembly of myofiber-specific PCG signatures was more consistent with the measurements of ATPase staining across 30 SMTs than the proportion of type I myofibers determined using only the *MYH7* isoform marker (Spearman's $r = 0.26$ with $P = 0.16$ for the *MYH7* marker vs. $r = 0.47$ with $P = 9.42 \times 10^{-3}$ for multiple PCG signatures). (**main text, lines 440-447**)

We appreciated this valuable suggestion, which further inspired us to investigate spatial resolution of individual myofibers in different species (**Figure R2**). We found that the cross-sectional area of myofibers spans from $736.42 \mu\text{m}^2$ in duck (~1/3 of the Visium spot area) to $9089.89 \mu\text{m}^2$ in frog (~ 4-fold that of the spot area), which suggested differences in the applicability of ST for generation of spatially resolved gene expression maps for SMTs across species. From this perspective, we can further improve our methods for future studies.

Figure R2. Statistics of cross-sectional area of myofiber in 11 species. Numbers at the top of the plot indicate the median cross-sectional area of myofibers. The red line indicates the Visium spot area. The percentage of myofibers with an area exceeding the spot area are shown above the error bar. The spot area was calculated as $\pi \times r^2$, where r is the spot radius.

Comment 1-3:

2. The authors use human data to provide a reference for cell type deconvolution in pig. While the authors perform this analysis carefully, using marker genes, the expression of these markers may very well have nevertheless changed in the orthologous porcine cell type. One strategy to make sure this is not a major issue is to analyze the robustness to differences in which markers are used in the deconvolution analysis. This analysis could simply involve picking a random subset of markers $X\%$ of the time (similar to a bootstrap analysis), and use this to provide a confidence range of cell type proportions. This type of analysis would strengthen the authors' points by demonstrating that major results are not driven by a single marker (or a small collection) that has changed expression in pig compared with human.

Response 1-3:

We very much appreciate this well-considered concern. To strengthen our estimations of relative cell-type proportions in adipose tissues using human signature matrix as a reference, we performed a bootstrap analysis, as suggested. As shown in **revised Supplementary Fig. 27**, we now present the estimated proportions accompanied by a confidence interval to account for these inherent uncertainties.

Revised Supplementary Fig. 27. Estimates of the proportions of different cell types

in ATs. Percentages next to pie plots show the proportion of each cell type, including adipocytes, microvascular endothelial cells (MVECs), macrophages (M1&M2 combined), and CD4+ T-cells. We also calculated the confidence range for cell type proportions using a bootstrap (1000 times) method that selects a random subset of 80% of the markers. The percentages in parentheses denote the confidence range for each estimated percentage.

Comment 1-4:

3. The authors conduct an interesting comparative analysis of enhancers in five species. The authors used data from several new in situ Hi-C libraries to estimate enhancer-promoter interactions. It's not clear to me (from either the results or methods sections) where the location of enhancers were obtained from. Please clarify where enhancers were obtained from.

Response 1-4:

PSYCHIC was applied to identify over-represented enhancers that interact with promoters. To this end, we examined a large genomic region surrounding each promoter (± 10 Mb) and searched for regions (*i.e.*, 20 Kb bins) enriched for Hi-C interactions with the promoters, which we then defined as the enhancers.

According to your suggestion, we provided statistical information for the PEIs and systematically characterized the features of the enhancers (**Supplementary Table 9**). The enhancers identified in this study exhibited similar characteristics to canonical enhancers, including their genomic distribution, sequence conservation, proximity rank of genes associated with a given enhancer, and preferential intra-TAD interactions.

We hope that this enhanced analysis will give the reader a better understanding of our comparative analysis of PEIs.

Supplementary Table 9. Number of PEIs in each species.

Species	Replicates	Number of PEIs	Enhancers per gene/promoter
Human	1	54,455	3.78
	2	55,133	3.93
	3	56,343	3.81
Rat	1	54,856	3.91
	2	56,746	3.93
Rabbit	1	54,898	4.73
	2	55,649	4.71
Sheep	1	57,817	3.69
	2	56,841	3.70
	3	56,104	3.58
Cat	1	57,507	3.97
Dog	1	55,981	3.86
	2	53,714	3.81
Pig	1	53,148	3.77
	2	55,960	3.89
	3	55,429	3.83
	4	55,710	3.87
	5	42,652	3.39
	6	55,733	3.87

Newly added Supplementary Fig. 38. Characteristics of promoter–enhancer interactions (PEIs).

a, Distribution (blue bars) and cumulative distribution (orange dots) of PEIs for each species, as predicted using PSYCHIC. Overall, ~37.27% (33.92% in human to 40.91% in dog) of the enhancers were located within 120 Kb of their target promoters.

b, Distribution of enhancer sequences across genomic features. The y-axis shows the percentage of total enhancers overlapping with different genomic features. These enhancers tend to overlap with intron and intergenic regions.

c, Conservation of enhancers in vertebrates. The x-axis depicts the start and end of enhancers flanked by 1 Mb of adjacent sequence. The y-axis represents sequence conservation calculated by 100-vertebrate phastCons or phyloP values. Enhancers showed higher sequence conservation than their immediate flanking regions,

d, Distribution (blue bars) and cumulative distribution (orange dots) of the proximity rank of genes associated with enhancer bins. Only about 10.77% (~8.53% in rat to 13.14% in rabbit) of enhancers regulate the nearest gene.

e, Percentage of PEIs within TADs. Black dots show percent of “random” enhancers residing within the same TAD. As expected, most (~68.02% in rat to 80.45% in sheep) of the detected interactions fall within the topologically associating domains (TADs), compared to 57.70% in random shuffles.

Reviewer #2**Comment 2-1:**

The manuscript by Jin et al. entitled "A Pig BodyMap Transcriptome Reveals Diverse Tissue Physiologies and Evolutionary Dynamics of Transcription" is data and information rich. After the BodyMaps of the human, mouse, and rat, the pig BodyMap reported here debuted large numbers of TUCP and CircRNA, together with large numbers of new miRNAs and lncRNAs.

The study should serve as an important reference for the scientific communities in both agriculture and biomedical research. In this regards, it has a wide readership and should be published in a high impact journal. The manuscript, however, is not written to the standards of Nature communication, albeit mostly understandably and logically presented. There are numerous grammatical errors, redundancies, long sentences coupled with inappropriate and misplaced prepositions. The authors should fully spell out all tissues names. The acronyms are unusual and together with grammatically problems, the text is very difficult to read.

Response 2-1:

We very much appreciate the reviewer's careful consideration of our study and we strive to provide the most clear and coherent description of our findings. To address vagaries in the language, we enlisted the services of native English speakers with PhD training in the life sciences to assist with improving and polishing the full text of the manuscript and supplemental materials. We are always glad for the opportunity to improve the quality of our report and appreciate any specific comments regarding specific grammatical errors or problems with sentence structure to ensure that our narrative is clear and easy to follow for scientists from any life science discipline.

Comment 2-2:

As bar plots or heatmaps, the GO plots contain only p-values. This website, <https://bioc.ism.ac.jp/packages/3.7/bioc/vignettes/enrichplot/inst/doc/enrichplot.html#dot-plot>, employs enrichGO plots. This type of plot includes

both the p-values and the numbers of genes for each GO category and is therefore more informative. The authors should use this type of plot in at least some figures in this manuscript. Please commit that your analysis pipeline code are published on Github.

Response 2-2:

As suggested, we have revised the bar plots of functional enrichment analysis to the more informative enrichGO plots, which included 9 panels, comprising 6 supplementary figures, as listed below.

Supplementary Fig. 10b	Supplementary information, lines 305-319
Supplementary Fig. 20b-e	Supplementary information, lines 518-531
Supplementary Fig. 21e	Supplementary information, lines 535-552
Supplementary Fig. 22c	Supplementary information, lines 555-567
Supplementary Fig. 26b	Supplementary information, lines 615-628
Supplementary Fig. 30	Supplementary information, lines 669-672

The analysis pipeline code central to our main results has been uploaded to the GitHub.

Comment 2-3:

A main question is: the study collected multiple tissues of all three germ layers. Are there any commonalities in transcription among tissues from the same germ layer? And similarly, what are the main differences among different germ layers?

Response 2-3:

We appreciate this interesting and constructive question. As suggested, we analyzed the commonalties and differences in transcription among tissues derived from different germ layers and added the new content to the main text (**page 7, lines 204-219**), and newly added **Supplementary Fig. 12**.

“Supporting the notion that the transcriptional programs may store memory of lineage specification and differentiation²⁷, we observed that the transcriptomic divergence was relatively higher between tissues originating from three different germ layers than that between tissues originating from the

same germ layer (**Supplementary Fig. 12a**). We observed that this profound difference was further amplified for 36 paralogous transcription factors (TFs) in the homeobox (*HOX*) superfamily, which are major regulators of animal morphogenesis and development (**Supplementary Fig. 12b**). We identified sets of germ layer-specific markers (2,507 for ectoderm, 1,053 for endoderm, and 874 for mesoderm) (**Supplementary Fig. 12c**) that were critical specifiers for different germ layers. For example, neurodevelopmental genes (such as *HK1*²⁸, *VWC2*²⁹, and *NECTIN1*³⁰), a metabolic gene (*FOXA3*)³¹, and genes involved in cardiovascular differentiation (*TMEM88*³² and *VIM*³³) were highly expressed in tissues derived from ectoderm (*i.e.*, cerebrum, cerebellum and retina), endoderm (*i.e.*, liver, small intestine, colon, and stomach), and mesoderm (typically, heart), respectively (**Supplementary Fig. 12d**).”

Newly added Supplementary Fig. 12. Expression profiling of tissues derived from three germ layers.

a, Heatmap (top) and boxplot (bottom) comparisons of transcriptional profiles for each of the five types of transcripts in tissues derived from distinct germ layers using Pearson's correlation. All 31 tissues were partitioned into 3 distinct germ layer-related groups: 3 ectoderm-derived tissues (cerebrum, cerebellum, and retina), 4 endoderm-derived tissues (colon, intestine, liver, and stomach), and 24 mostly mesoderm-derived tissues (*e.g.*, heart, lung, and uterus *etc.*), listed in **Supplementary Table 1**, as described in Hon *et al.*⁶

b, Transcriptional patterns of *HOX* genes suggesting the developmental origins of these tissues by germ layer. A randomly selected equal number of PCGs were used as the

control (grey box) and showed higher coefficients than *HOX* genes (Wilcoxon rank-sum test).

c, Heatmap with the expression of identified marker genes for the definitive germ layers (2,507 for ectoderm, 1,053 for endoderm, and 874 for mesoderm). Each gene's normalized log expression levels are standardized so that they range between 0 and 1.

d, Bar plot showing the expression patterns of 6 marker genes from (c).

Comment 2-4:

Fig. 1: the term “transcribed transcript” is very confusing. Are there un-transcribed transcripts? In Table 1, “detected transcripts” may be a better term. For the rest of the text, simply “transcript” is sufficient.

Response 2-4:

As suggested, we have changed "transcribed transcript" to "detected transcripts".

Comment 2-5:

Fig 2. If the study includes 20 chromosomes for the pig (from Sup Fig. 1), why are X and Y not in this figure?

Response 2-5:

In this study, we used female pigs to construct the 3D genomic structure in adipose tissue. X chromosome inactivation in females leads to the loss of local structure and subsequent formation of a bipartite structure with two superdomains².

This overriding effect of X-chromosome inactivation causes an artifactual result in Hi-C maps for X chromosomes. Since we therefore could not determine with certainty that 3D structures of X chromosome were *bona fide*, we opted to exclude X chromosomes from the analysis.

Correspondingly, we have added some description about this into the legend of **Fig. 2**: "X chromosome is excluded from this analysis due to the overriding effect of X-chromosome inactivation".

Comment 2-6:

L154-155: Are these TPM values regarded as high, medium, low? Please add in Table 1 the ranges of TPMs for PCGs and the other types of transcripts.

Response 2-6:

As suggested, we have added **Supplementary Fig. 9**, which describes the distribution of transcriptional abundances of five types of transcripts.

Newly added Supplementary Fig. 9. Expression distribution of five types of transcripts. The x-axis indicates the transcripts sorted from lowest to highest abundance. Colored lines represent mean values across tissues, and lighter-colored shading around the mean represents dispersion, calculated as the standard deviation divided by the cumulative sum of all means. The red vertical dashed lines indicate median expression level. The left and right blue vertical dashed lines indicated 25th and 75th quantiles, respectively. The upper box plot also shows the expression distribution.

Comment 2-7:

L162-166: this statement is more than 4 lines' long. It is also written in a very convoluted and confusing fashion. Please revise.

Response 2-7:

As suggested, we have revised this sentence (**main text: page 6, lines 175-179**).

“The most abundant transcripts (the top 0.5%) in a tissue, accounted for greater than half of the total transcribed TUCPs (~58.35%), lncRNAs (~75.53%), and miRNAs (~62.34%), whereas PCGs (~46.26%) and circRNAs (~35.48%) had a more uniform distribution.”

Comment 2-8:

L170: what is the “x” for?

Response 2-8:

It was a typographical error, which we have corrected.

Comment 2-9:

L192: what are these “individuals”? They are all from one pig.

Response 2-9:

In this context, ‘individuals’ indicated different biological replicates. We have changed ‘individuals’ to ‘pigs’.

Comment 2-10:

L209-210: statement not finished.

Response 2-10:

We have reformatted the statement, from “In contrast, compartment B (1,161 Mb in length) regions were GC-poor (38.26%), transcript-sparse (4.23 PCGs per Mb) and poorly transcribed (0.18 TPM for PCGs), and tended toward the periphery.”

to

“In contrast, compartment B (1,161 Mb in length) regions were GC-poor (38.26%), transcript-sparse (4.22 PCGs per Mb) and poorly transcribed (0.33 TPM for PCGs), and also tended to be localized in the periphery.”

Comment 2-11:

L244: axis of ?

Response 2-11:

The term “anterior-posterior axis” indicates a line running from the head to the

tail end of a bilaterally symmetrical organism.

We have revised the description from “anterior/posterior axis formation” to “**formation of the anterior-posterior axis (i.e., from head to hindlimb)**”. (main text: **page 10, lines 282-283**)

Comment 2-12:

L274: Only “hundreds of SMTs are controlled by the nervous system”? My understanding is that all of them are. Please check this.

Response 2-12:

Thank you for pointing this out. We have deleted ‘hundreds of’.

Comment 2-13:

L304-309: another long and confusing sentence that needs re-written.

Response 2-13:

We have re-written this sentence. (main text: **page 12, lines 351-358**)

“**We performed clustering analysis to compare expression patterns of ten annotated MYH isoforms across SMTs (Supplementary Fig. 19a) and found that they were weakly consistent with the global transcription patterns of PCGs and other transcript types (cophenetic $r = 0.343$ for PCGs, 0.328 for TUCPs, 0.329 for lncRNAs, 0.060 for circRNAs, and 0.138 for miRNAs).**”

Comment 2-14:

L315-319: again long and poorly written sentence.

Response 2-14:

We have re-written this sentence. (main text: **page 12, lines 366-371**)

“**We found that changes in the abundance of the type I, IIA, and IIB marker isoforms were positively correlated with global divergence in the transcriptome (reflected by the proportions of each of the five transcript types that showed**

altered expression in pair-wise comparisons of SMTs).”

Comment 2-15:

L328: the term “myofiber-specific associated” is very confusing. If the transcripts are myofiber-specific, why add “associated”?

Response 2-15:

We have deleted “associated”.

Comment 2-16:

L551: instead of calling the chicken a “group”, “an evolutionally distant species” may be more appropriate.

Response 2-16:

Since outgroup is a standard term in phylogenetic analyses of evolutionary relationships that serves as a reference point for the scale among hypothetically more closely related organisms, we prefer to use this term. However, we understand the need for clarity for all readers and we thus use “**an evolutionarily distant outgroup**” to indicate its inclusion as an evolutionary reference point. (main text: page 21, line 648)

Comment 2-17:

L625-626: Please use short phrases to explain “phastCons values” and “phyloP values”. It will also help the reader if you explain why they imply that the divergence of PCG transcription across species is independent of sequence conservation.

Response 2-17:

Thank you for raising this valuable point. As suggested, we have revised the corresponding text from “**PCGs with a high CV (that were thus evolutionarily variable) exhibited marginal differences in phastCons values (0.13 vs. 0.11, *P***

= 0.054) and phyloP values (0.16 vs. 0.13, $P = 0.029$)” to “PCGs with a high CV (*i.e.*, evolutionarily variable) exhibited only marginal differences in nucleotide sequence conservation estimated by phastCons values (0.11 vs. 0.13, $P = 0.063$) and phyloP values (0.13 vs. 0.21, $P = 0.188$) (Fig. 7d, Supplementary Fig. 40)”. (main text: page 24, lines 729-732)

Comment 2-18:

L641: somatic cell nuclear transfer is the formal scientific term for cloning.

Response 2-18:

As suggested, we have changed "somatic nuclear cloning" to "somatic cell nuclear transfer". (main text: page 25, line 750)

Comment 2-19:

L680, 687: "in pig" is not correct. When referring to a species, it should be either "in pigs" or "in the pig".

Response 2-19:

Changed as suggested.

Comment 2-20:

L881-883: For the three breeds of Chinese pigs used, please provide references that describe their physiological/production characteristics and genetic backgrounds.

Response 2-20:

Thanks for this suggestion. We have provided descriptions of the phenotypic characteristics of each pig breed used in this study in corresponding revisions in the Supplementary Methods. (page 81, lines 992-998)

“More specifically, most of these samples were collected from three adult 2-year-old female Rongchang pigs (a fatty, indigenous Chinese breed, white in

colour, with black spots on the head, and drooped ears), while the testes were from three 2-year-old male Large White pigs (a lean, commercial European breed, white in colour with erect ears) and retinal tissues were derived from two 2-day-old female crossbred Meishan (father; a fatty, indigenous Chinese breed, black in colour with drooped ears) × Tibetan (mother; an indigenous Chinese breed, black in colour with erect ears) pigs.”

Comment 2-21:

Sup Fig. 2. No figure legend.

Response 2-21:

Thank you for pointing out this inadvertent omission. We have added the figure legend, accordingly, and also used colors to distinguish separate phases in the analytical workflow.

Revised Supplementary Fig. 2. Workflow for reconstruction of the pig transcriptome.

Sequence reads were aligned to the pig reference genome (Sscrofa 11.1, GCA_000003025.6) by the STAR alignment tool (version 2.5.3a). The aligned reads of these samples were assembled using Cufflinks (version 2.1.1). We then filtered out library-specific background noise and predicted the most likely isoforms from the assemblies of transcript fragments (transfrags) (see more details in Supplementary Methods). After filtering, the remaining high-quality transcript assemblies were then subjected to TACO, which led to construction of transcriptome maps. We then assessed the coding potential of putative non-coding transcripts that were not annotated as PCGs in the pig reference

genome by integrating two sources of evidence: coding potential calculator and detected Pfam A domain matches. Transcripts without coding potential (CPC score <0 and without Pfam domain hits) were defined as lncRNAs, otherwise TUCPs.

Unmapped reads from the STAR mapping were retrieved and used for circRNA prediction using the CIRCEplorer2.

The small RNA-seq data were mapped to the reference genome using Bowtie. Mappable reads were submitted to miRDeep (version 2.0.0.7) to detect miRNAs, while annotated mature miRNA sequences of pig and all other mammalian and avian species in miRbase (release 22) were selected as references.

Comment 2-22:

Sup Fig. 4a. Red and yellow could not be distinguished, change the color of one of them to a different color, black, for example.

Response 2-22:

Changed as suggested.

Comment 2-23:

Sup Fig. 5a. Please explain what the different colored lines are. If they denote genomic DNA, why do they have different colors? Also, "intergenic lncRNA" is used in Fig.5a, while in Fig. 5b. it is changed to LincRNA. If these are the same, please be consistent. Fig. 5b. There should be a space separating a bracket from the preceding text. This is a recurring issue in the entire supplementary file. Please change them all.

Response 2-23:

Yes, you are correct. Thanks for your careful attention to detail. The lines denote genomic DNA and should be consistent in color. We sincerely apologize for these format issues. We have made corresponding revisions and carefully checked all analogous issues across the entire supplementary file.

Comment 2-24:

Sup Fig. 6a and b. The color keys in 6a are not the same as those in the diagram. Both a and b should be made three dimensional with multiple X-axes so the data can be better discerned.

Fig. 6e: please explain the color scale, density of ? Is this a log scale?

Response 2-24:

As suggested, we have redrawn **Supplementary Fig. 7a-b** using colored lines to make the data more clearly discernible.

The color bar in **Supplementary Fig. 7e** denotes the \log_{10} -transformed two-dimensional kernel density. We have added a statement to the figure legend to better explain this point.

Comment 2-25:

Sup Fig. 7a. % of Relative abundance of circRNA in ? in total? 7c: please explain Q1-Q4 here. What do these TPM values imply? If TPM>0.1 is regarded as transcribed, are these still transcribed?

Response 2-25:

We apologize for the unclear labeling. The relative abundance of circRNA indicates the percentage of circular junction reads from among all reads mapped to the genome. We have added the following description to the figure legend.

“a, Relative abundance of circRNA reads. The percentage of circular junction reads from all reads mapped to the reference genome is shown for different tissues. The neural tissues (especially retina and cerebellum) have prominently more circRNA reads than other tissues, while the two cell lines have the fewest circRNA reads.” (page 14, lines 272-273)

Q1-Q4 here indicate in the four quartiles of expression levels, which were used to eliminate the possibility that the features of circRNAs, such as back-spliced exon positions, length of introns flanking circularized exons, number of

exons, or exon length, etc. were affected by the level of circRNA expression. The detected circRNAs were defined by having TPM >0.05 in at least one sample, as published in a previous study³. These detected circRNAs were then divided into the four quartiles of expression levels containing equal number of circRNAs. We observed similar patterns in all of the expression intervals (*i.e.*, more circRNAs were generally produced from the middle exons of transcripts). We have also accordingly revised the statement in the figure legend as follows:

“This result is presented in four quartiles of expression levels from Q1 to Q4 (each containing an equal number of circRNAs) which indicate TPM values from low to high. Q1: 0.05-0.06 TPM, Q2: 0.06-0.083 TPM, Q3: 0.083-0.15 TPM, and Q4: >0.15 TPM.” (page 14, lines 282-284)

Comment 2-26:

Sup Fig. 12b-e: Please add the color keys.

Response 2-26:

Revised as suggested.

Comment 2-27:

Sup Fig. 16b-f: please explain the color scales and the dots in the plots.

The density scales on the X- and Y-axes are in decimal and percentage, respectively. Shouldn't they be in one consistent format?

Response 2-27:

We have added a description of the color scales into the figure legend.

We have revised the scales for the X- and Y-axes so that they are in a consistent format, which indicates frequency.

Comment 2-28:

L319: “between SMTs (Supplementary Fig. 16 b-f)”. These figures do not show which pairs of SMTs were compared, neither in the plots nor in the legend.

Response 2-28:

Thank you for pointing out this inadequate description. This figure indicated all pair-wise comparisons of SMTs. We have revised the text in the legend as follows:

“The x- and y-axes indicate the DE gene ratio and the log₂-transformed *MYH* transcription difference of all pairwise SMT comparisons, respectively.” (page 27, lines 502-503)

Comment 2-29:

Sup Fig. 22. Both panels of the figure are very small and it is not possible to see the GO terms in b.

What do the color shades indicate on the X- and Y-axes in b?

Response 2-29:

According to your suggestion, we have redrawn **Supplementary Fig. 26b** in a more clear and informative format, so that it now includes both the *P*-values, as well as the number and ratio of genes. We also added a description explaining the color shades into the figure legend.

Revised Supplementary Fig. 26. Transcriptional divergence of distinct transcript types across adipose tissues (ATs).

a, Multidimensional scaling through t-SNE distances based on \log_2 -transformed transcription levels of different transcript types.

b, Functional enrichment for differentially expressed (DEs) PCGs between ATs. Plot showing the top 10 enriched GO terms of upregulated DE PCGs in pairwise AT comparisons. The pairwise comparisons listed at the bottom are sorted as follows: between SATs and VATs, within SATs, and within VATs. The names highlighted in red for each

pairwise comparison represent the tissue in which the genes were highly expressed. The color shades on the Y-axes highlight the categories of GO terms, *i.e.*, extracellular matrix organization related terms (orange), inflammation and immunity related terms (blue), and others (grey). The sizes of dots represent numbers of enriched genes, and dot color represents the $-\log_{10}(P\text{-value})$. Heights of lines in each square indicate the ratio of enriched genes.

Comment 2-30:

Sup Fig. 24: please explain the Z-score.

Response 2-30:

We have added description explaining the Z-scores to the figure legend. Accordingly, we have also checked other figures and added description to the legends where appropriate.

“The expression levels were standardized by Z-score (mean of zero and s.d. of one) for each gene. The color bar indicates Z-score range of expression level, calculated as standard deviation from average expression level divided by standard deviation.” (page 38, lines 647-650)

Comment 2-31:

Sup Fig. 28a: Please provide the color keys for the samples (lanes).

Response 2-31:

Revised as suggested.

Comment 2-32:

Sup Fig. 33. Please add a statement summarizing the data presented.

Response 2-32:

We have added the following statement to the figure legend:

“**Supplementary Fig. 37. Gene transcription phylogenies for 7 tissues across 9 mammals.** We reconstructed neighbor-joining tree based on distances ($1-r$, here, r is Spearman’s correlation coefficient) between

transcription levels of single-copy orthologs. The total branch lengths of each tree were extracted to estimate the evolutionary divergence of transcription profiles between tissues, as presented in **Fig. 6g**. The scale bar represents distances.”

Reviewer #3:**Comment 3-1:**

The authors present a detailed analysis of an extensive pig transcriptome dataset. The manuscript is largely descriptive but nevertheless is comprehensive and very thorough. These data and the associated analyses represent a very valuable resource for researchers working on pigs both as biomedical models and as an important agricultural animal.

Response 3-1:

We are very grateful for the reviewer's careful consideration and positive remarks in support of our study.

Comment 3-2:

The major issue that needs to be addressed prior to acceptance is the availability of the data. The authors cite accession numbers for NCBI (PRJNA637678 and PRJNA655361) and National Genomics Data Center of China (<http://bigd.big.ac.cn/gsa-human>) (PRJCA003737) but searches of these data repositories with these accession numbers return no results.

Response 3-2:

We appreciate your reminder to ensure the public accessibility of our data; we are keenly and profoundly aware that our study represents a resource for the research community as a whole. We have uploaded all of the data described in the manuscript to the NCBI Gene Expression Omnibus (GEO) database. Accordingly, we have also updated the Data availability statement.

Data availability

The authors declare that all data supporting the findings of this study are available within the article and its Supplementary Information files or from the corresponding author upon reasonable request.

Raw and processed RNA, and miRNA sequencing data of pig have been deposited in the NCBI Gene Expression Omnibus (GEO) under accession

codes GSE162145 and GSE162147, respectively. Spatial transcriptomic data have been deposited in GEO under accession code GSE161882. The raw and processed RNA sequencing data of nine other species for comparative transcriptomic analysis have been deposited in GEO under accession code GSE162142. RNA-sequencing and Hi-C data of non-human species for analysis of gene transcription divergence and PEIs across species have been deposited in GEO under accession codes GSE162146 and GSE162140, respectively. Human RNA sequencing and Hi-C data have been deposited in GEO under accession codes GSE162143 and GSE162139, respectively.

We also provided reviewer access as below.

To review GEO accession GSE162145:

Go to <https://www.ncbi.nlm.nih.gov/geo/query/acc.cgi?acc=GSE162145>

Enter token qjolgawfjupvaj into the box

To review GEO accession GSE162147:

Go to <https://www.ncbi.nlm.nih.gov/geo/query/acc.cgi?acc=GSE162147>

Enter token enchqaecxfujnqd into the box

To review GEO accession GSE161882:

Go to <https://www.ncbi.nlm.nih.gov/geo/query/acc.cgi?acc=GSE161882>

Enter token kdatquosvtujnub into the box

To review GEO accession GSE162142:

Go to <https://www.ncbi.nlm.nih.gov/geo/query/acc.cgi?acc=GSE162142>

Enter token ircbumigxpqxriz into the box

To review GEO accession GSE162146:

Go to <https://www.ncbi.nlm.nih.gov/geo/query/acc.cgi?acc=GSE162146>

Enter token kvqfeciqlgpxul into the box

To review GEO accession GSE162140:

Go to <https://www.ncbi.nlm.nih.gov/geo/query/acc.cgi?acc=GSE162140>

Enter token qbgrmygeznqdbj into the box

To review GEO accession GSE162143:

Go to <https://www.ncbi.nlm.nih.gov/geo/query/acc.cgi?acc=GSE162143>

Enter token sxitoagmpxsspot into the box

To review GEO accession GSE162139:

Go to <https://www.ncbi.nlm.nih.gov/geo/query/acc.cgi?acc=GSE162139>

Enter token kvqloskkfpqnvql into the box

Comment 3-3:

The following would be more appropriate references in relation to the FAANG project (line 68):

*Andersson L, Archibald AL, Bottema CD, Brauning R, Burgess SC, Burt DW, Casas E, Cheng HH, Clarke L, Couldrey C, Dalrymple BP, Elsik CG, Foissac S, Giuffra E, Groenen MA, Hayes BJ, Huang LS, Khatib H, Kijas JW, Kim H, Lunney JK, McCarthy FM, McEwan JC, Moore S, Nanduri B, Notredame C, Palti Y, Plastow GS, Reecy JM, Rohrer GA, Sarropoulou E, Schmidt CJ, Silverstein J, Tellam RL, Tixier-Boichard M, Tosser-Klopp G, Tuggle CK, Vilkki J, White SN, Zhao S, Zhou H; FAANG Consortium. Coordinated international action to accelerate genome-to-phenome with FAANG, the Functional Annotation of Animal Genomes project. *Genome Biol.* 2015 Mar 25;16(1):57. doi: 10.1186/s13059-015-0622-4. PMID: 25854118; PMCID: PMC4373242.*

Foissac S, Djebali S, Munyard K, Vialaneix N, Rau A, Muret K, Esquerré D, Zytnicki M, Derrien T, Bardou P, Blanc F, Cabau C, Crisci E, Dhorne-Pollet S, Drouet F, Faraut T, Gonzalez I, Goubil A, Lacroix-Lamandé S, Laurent F, Marthey S, Marti-Marimon M, Momal-Leisenring R, Mompert F, Quéré P, Robelin D, Cristobal MS, Tosser-Klopp G, Vincent-Naulleau S, Fabre S, Pinard-Van der Laan MH, Klopp C, Tixier-Boichard M, Acloque H, Lagarrigue S, Giuffra

E. Multi-species annotation of transcriptome and chromatin structure in domesticated animals. BMC Biol. 2019 Dec 30;17(1):108. doi: 10.1186/s12915-019-0726-5. PMID: 31884969; PMCID: PMC6936065.

Response 3-3:

As suggested, we have added these references for the FAANG project.

Comment 3-4:

Lines 105-106:

It appears that the comparison with the Ensembl annotation of the Sscrofa11.1 reference genome that is highlighted in lines 105-106 concerns Ensembl Release 90, August 2017 that included only 361 annotated lncRNA genes. The paper by Warr et al. 2020 that describes the Sscrofa11.1 reference genome cited Ensembl Release 98, September 2018 that included annotation for 6,798 lncRNA genes.

Response 3-4:

Thanks for raising this point. The updated Ensembl annotation in the Sscrofa 11.1 reference genome greatly improved the annotation of lncRNAs, from only 361 lncRNA genes in Release 90 (August 2017) to 6,798 in Release 98 (September 2019), and to 6,797 in the latest Release 102 (November 2020).

We compared our lncRNA annotation to that in the latest version of the pig reference genome annotation (Release 102) and found that among the 19,072 lncRNAs, 12,180 (63.86%) of these lncRNAs have not yet been annotated in the reference genome. We also revised the description in the main text (**page 4, lines 112-113**).

Comment 3-5:

Lines 112-113

Similarly, it appears that the comparison with the Ensembl annotation of the Sscrofa11.1 reference genome that is highlighted in lines 112-113 concerns Ensembl Release 90, August 2017 that included 22,452 protein coding genes.

The paper by Warr et al. 2020 that describes the Sscrofa11.1 reference genome cited Ensembl Release 98, September 2018 that included annotation for 21,301 protein coding genes.

Response 3-5:

Yes, thanks for pointing out this issue in annotation as well. The continual updating of Ensembl annotation of the Sscrofa 11.1 reference genome has improved the annotation of protein coding genes (PCGs), resulting in changes from 22,452 PCGs in Release 90 (August 2017) to 21,301 in Release 98 (September 2019), and more recently to 21,303 in the latest Release 102 (November 2020).

Since the analyses in our original manuscript submission were done before the release of version 98, at that time we decided to present all of the results based on release 90, as we assumed that the main conclusions would not change by using the new version of the annotations. In the revised manuscript, we have now updated all of the results related to PCGs using the latest release 102 of annotation, which involved changes to 82 panels, throughout 31 figures, as well as 3 supplementary data files. We also updated ~176 statistical information in the main text. The new results are highly consistent with the original results with scarcely any differences, which generally supported our conclusions. For example, transcriptional profiles among tissues, using the new PCG quantification, were generally similar to the original version (**Figure R3**).

Figure R3. Transcriptional profiles across tissues based on the expression quantified by PCG annotation release 90 (a) and 102 (b).

Comment 3-6:

Regardless there is an inconsistency between the text in line 112 that refers to 22,452 PGC and Supplementary Figure 1 that cites 22,342 PGC.

Response 3-6:

Thank you for pointing this out. We have checked and revised this inconsistency. We have also corrected the annotation release version/time presented in the figure from 90 (August 2017) to 102 (November 2020).

Comment 3-7:

The authors included two pigs cell lines in their study kidney epithelial cells [PK15] and iliac endothelial cells [PIECs] but do not discuss to what extent these cells provide a good representation of the original source tissues, i.e. kidney and ileum.

Response 3-7:

In this study, we mainly aimed to construct a comprehensive pig transcriptome by *de novo* transcriptome assembly. In order to expand the richness of the source collections, we also included two widely used immortalized pig cell lines (PK15 and PIECs, the only two publicly available cell lines of which we are aware) which were respectively obtained from the China Infrastructure of Cell Line Resources and Stem Cell Bank of the Chinese Academy of Sciences.

We agree that the investigation of whether there is difference between epithelial cell lines from different tissues is a highly fascinating research topic. However, in our study, PIEC cell lines were derived from iliac artery, without matched origin tissue. We only included PK15 and its single origin tissue, kidney. Further investigation in future work will deepen our understanding of how epithelial cell lines derived from different organs may exhibit differences in their transcriptional profiles.

Comment 3-8:

It would be useful to state whether the cDNA / RNA-Seq libraries were oligodT or random primed.

Response 3-8:

As suggested, we added this information to the revised **Methods** section.

“We constructed rRNA-depleted and random priming RNA-seq libraries.” (main text: page 27, line 810)

Comment 3-9:

The following papers are also relevant to the characterization of the pig transcriptome:

*Li Y, Fang C, Fu Y, Hu A, Li C, Zou C, Li X, Zhao S, Zhang C, Li C. A survey of transcriptome complexity in *Sus scrofa* using single-molecule long-read sequencing. DNA Res. 2018 Aug 1;25(4):421-437. doi: 10.1093/dnares/dsy014. PMID: 29850846; PMCID: PMC6105124.*

Beiki H, Liu H, Huang J, Manchanda N, Nonneman D, Smith TPL, Reecy JM, Tuggle CK. Improved annotation of the domestic pig genome through integration of Iso-Seq and RNA-seq data. BMC Genomics. 2019 May 7;20(1):344. doi: 10.1186/s12864-019-5709-y. PMID: 31064321; PMCID: PMC6505119.

*Summers KM, Bush SJ, Wu C, Su AI, Muriuki C, Clark EL, Finlayson HA, Eory L, Waddell LA, Talbot R, Archibald AL, Hume DA. Functional Annotation of the Transcriptome of the Pig, *Sus scrofa*, Based Upon Network Analysis of an RNAseq Transcriptional Atlas. Front Genet. 2020 Feb 14;10:1355. doi: 10.3389/fgene.2019.01355. PMID: 32117413; PMCID: PMC7034361.*

Response 3-9:

Thank you for providing these three papers that are also highly relevant to characterization of the pig transcriptome. We appreciate the value of using different datasets with more annotated lncRNA/protein coding genes for comparative analyses. Unfortunately, although germane to the pig transcriptome, the annotation files of these papers were not accessible to us.

Thus, we compared lncRNAs in this study to other lncRNA datasets including the annotations (release 102) in pig reference genome, other published studies characterizing the pig transcriptome, as well as pig lncRNA collections in public

repositories, *i.e.*, the domestic-animal lncRNA database (ALDB) and the NONCODE database.

We assessed the sequence similarity and the exonic structural features of lncRNAs. We have added **Supplementary Fig. 6** describing the results of these comparative analyses.

Newly added Supplementary Fig. 6. Comparative analysis between lncRNAs in this study and other catalogs.

a, Pie charts of the number of lncRNA genes that are unannotated or annotated in pig reference genome (release v102).

b, Pie chart (left) showing the proportion of lncRNAs in this study that were previously discovered by only one or multiple (from 2 to 6) datasets listed in the table (right). About 36.38% were not discovered by any of the catalogs. Basic Local Alignment Search Tool (BLASTN) was used to discriminate the previously discovered lncRNAs in pig reference genome (v102)², Tang *et al.* (2017)³, Yang *et al.* (2017)⁴, Kern *et al.* (2018)⁵ and ALDB⁶, NONCODE database⁷ using the following criteria: e-value < 1×10^{-10} , min-identity of 80%, and min-coverage of 80%. **c**, Features (transcript length, lengths of exons, number of exons per transcript, and number of isoforms per gene) reported in different lncRNA catalogs.

References

- 1 Stahl, P. L. *et al.* Visualization and analysis of gene expression in tissue sections by spatial transcriptomics. *Science* **353**, 78-82 (2016).
- 2 Giorgetti, L. *et al.* Structural organization of the inactive X chromosome in the mouse. *Nature* **535**, 575-579 (2016).
- 3 Veno, M. T. *et al.* Spatio-temporal regulation of circular RNA expression during porcine embryonic brain development. *Genome Biol.* **16**, 245 (2015).

Reviewers' Comments:

Reviewer #1:

Remarks to the Author:

It looks like the authors have addressed my comments.

Reviewer #2:

Remarks to the Author:

The authors are commended for making substantial revisions/changes based on my comments. I consider my comments of scientific nature fully addressed. The writing, however, still needs editing. For example, in the first round of review, I commented that when referring to a species, either "the pig", or "pigs" should be used. But the authors changed "the pig" to "pig" (multiple locations, see L33,52, 94, 116). This makes no sense. The entire manuscript needs to be checked for this erroneous change. Other similar issues such as "rat" (L72) are still not corrected. Another example is the clunky sentence structure such as "between tissues originating from three different germ layers than that between tissues originating from the same germ layer". This can simply be "between tissues of different germ layers than within germ layers". There are also numerous grammatical mistakes in the text.

It would be a pity if such good data are not accompanied with the best writing.

Reviewer #3:

Remarks to the Author:

This manuscript describes a thorough analysis of transcriptomes of multiple tissues from the pig and includes comparisons with transcriptomes from homologous tissues in some other vertebrate species. These data represent an invaluable resource to scientists engaged in research on pigs in both agricultural or biomedical contexts. The manuscript has been improved by the revisions made by the authors.

The underlying sequence data are in the GEO public database as stated by the authors. It is disappointing that these data are embargoed until November 2023. It is hoped that this two year long embargo represents the authors unfounded pessimism about how long it will take for this manuscript to be accepted for publication and published. It is hoped that the embargo will be lifted as soon as this manuscript is published. The genomics community has encouraged PRE-publication data release under the terms of the Fort Lauderdale Agreement and as confirmed in the Toronto Statement (Toronto International Data Release Workshop. Birney et al. 2009. Pre-publication data sharing. Nature 461:168-170). These agreements facilitate early sharing of data whilst protecting the rights of the data generators to first genome-wide analysis and publication.

The following minor changes to the manuscript are suggested:

Line numbers in these comments refer to those in the marked up copy of the manuscript (283705_1_art_file_5381151_qph5pr.pdf),

Line 33: ".of the pig.." – this maybe a matter for the copy editor grammatically the 'the' should be retained.

Line 51-52: the following would be better english "Comparative analysis of the transcriptomes of seven tissues from pigs and nine other vertebrates revealed evolutionary divergence in transcription that potentially contributes to lineage-specific biology."

Line 76: (Sscrofa11.1)11 – reference 10 does not report the new pig genome assembly.

Line 90: change "the construction an atlas..." to "the construction of an atlas..."

Lines 175-179: This sentence is somewhat difficult to understand. Changing the start of the sentence to "The most abundantly expressed transcripts (the top 0.5%) in a tissue,....." or "The most abundant transcripts (the top 0.5%, as ranked by expression levels) in a tissue,..."

Line 264: "funcitons" to "functions"

Lines 642-649: the following change would improve the english of this sentence, change "..we performed a comparative transcriptomic analysis of seven homologous tissues from pig and nine representative vertebrate mammalian models including..." to "we performed comparative analyses of the transcriptomes of seven homologous tissues from pigs and nine representative vertebrate mammalian models including..."

Line 678: change "(the most closely rodent models,...)" to "(the most closely related rodent models,...)"

Lines 681-682: change "than it was to other non-rodent..." to "than to other non-rodent..."

Lines 736-741: it would be appropriate to cite a reference for this hypothesis.

Line 758: change "economically important products of pig." to "economically important products."

Line 760: change "pig" to "pigs"

Line 790: it may be worth adding a topical note that human ACE2 has been shown to act as a receptor for SARS-CoV-2

Line 795: change "pig" to "pigs"

Line 926: change "Ensemble" to "Ensembl"

Figure 1c suggests that the correlation in expression patterns between a) PK15 and kidney and b) PEIC and small intestine or colon are 0.8 or less and that these two cell lines have expression patterns more similar to ovary and uterus than the tissues from which these cell lines were developed. These comparisons are worth some comment in the manuscript.

A footnote to Table 1 should be added to confirm which annotation of Sscrofa11.1 is being cited. Based on the number of lncRNA cited (i.e. 6,797) it appears to be Ensembl release 100, April 2020, Ensembl release 101, August 2020 or Ensembl release 102 as stated in the rebuttal letter. However, in all of these releases the number of PCGs is 21,303, not 21,280.

Detailed responses to reviewers

All comments provided by reviewers are in gray italics, and our responses are in black. Important revisions in the manuscript are marked in red.

Reviewer #1

Comment 1:

It looks like the authors have addressed my comments.

Response 1:

We sincerely appreciate your comments and advice for improving the manuscript.

Reviewer #2

Comment 2-1:

The authors are commended for making substantial revisions/changes based on my comments. I consider my comments of scientific nature fully addressed.

Response 2-1:

We sincerely appreciate your comments and advice for improving the manuscript.

Comment 2-2:

The writing, however, still needs editing. For example, in the first round of review, I commented that when referring to a species, either “the pig”, or “pigs” should be used. But the authors changed “the pig” to “pig” (multiple locations, see L33,52, 94, 116). This makes no sense. The entire manuscript needs to be checked for this erroneous change. Other similar issues such as “rat” (L72) are still not corrected. Another example is the clunky sentence structure such as “between tissues originating from three different germ layers than that between tissues originating from the same germ layer”. This can simply be “between tissues of different germ layers than within germ layers”. There are also numerous grammatical mistakes in the text.

It would be a pity if such good data are not accompanied with the best writing.

Response 2-2:

To address vagaries in the language, we enlisted the services of a native English speaker who is a certified editor in the life sciences (ELS) with PhD training to assist with improvement and polishing of the full manuscript and supplemental materials.

Regarding the use of “the pig” or “pigs” to refer to a species in general or individual animals, we have corrected these descriptions so that they are grammatically correct and also checked for similar issues throughout the manuscript.

As suggested, we have simplified the sentence (**main text: page 6, lines 180-181**).

“...we observed that the transcriptomic divergence was relatively higher between tissues of different germ layers than within germ layers.”

Thank you for your advice.

Reviewer #3:**Comment 3-1:**

This manuscript describes a thorough analysis of transcriptomes of multiple tissues from the pig and includes comparisons with transcriptomes from homologous tissues in some other vertebrate species. These data represent an invaluable resource to scientists engaged in research on pigs in both agricultural or biomedical contexts. The manuscript has been improved by the revisions made by the authors.

Response 3-1:

We sincerely appreciate your comments in improving the manuscript.

Comment 3-2:

The underlying sequence data are in the GEO public database as stated by the authors. It is disappointing that these data are embargoed until November 2023. It is hoped that this two year long embargo represents the authors unfounded pessimism about how long it will take for this manuscript to be accepted for publication and published. It is hoped that the embargo will be lifted as soon as this manuscript is published. The genomics community has encouraged PRE-publication data release under the terms of the Fort Lauderdale Agreement and as confirmed in the Toronto Statement (Toronto International Data Release Workshop. Birney et al. 2009. Pre-publication data sharing. Nature 461:168-170). These agreements facilitate early sharing of data whilst protecting the rights of the data generators to first genome-wide analysis and publication.

Response 3-2:

We appreciate your relatively (and unexpectedly) rapid review of our work. To ensure that our data are widely available and accessible to the research community, we have now made the data publicly available.

Comment 3-3:

The following minor changes to the manuscript are suggested: Line numbers in these comments refer to those in the marked up copy of the manuscript (283705_1_art_file_5381151_qph5pr.pdf),

Response 3-3:

We are very grateful for the reviewer's careful attention to detail and, as suggested, we have made revisions based on the specific comments listed below.

Comment 3-4:

Line 33: “..of the pig..” – this maybe a matter for the copy editor grammatically the ‘the’ should be retained.

Response 3-4:

Thank you for pointing this out. We have changed it to “of pigs”.

Comment 3-5:

Line 51-52: the following would be better english “Comparative analysis of the transcriptomes of seven tissues from pigs and nine other vertebrates revealed evolutionary divergence in transcription that potentially contributes to lineage-specific biology.”

Response 3-5:

Changed as suggested. (**main text: page 2, lines 40-43**)

Comment 3-6:

Line 76: (Sscrofa11.1)¹¹ – reference 10 does not report the new pig genome assembly.

Response 3-6:

Thanks for your careful attention to detail. We have deleted reference 10, as appropriate.

Comment 3-7:

Line 90: change “the construction an atlas...” to “the construction of an atlas...”

Response 3-7:

Changed as suggested. (main text: page 3, line 73)

Comment 3-8:

Lines 175-179: This sentence is somewhat difficult to understand. Changing the start of the sentence to “The most abundantly expressed transcripts (the top 0.5%) in a tissue,.....”

or “The most abundant transcripts (the top 0.5%, as ranked by expression levels) in a tissue,...”

Response 3-8:

Changed as suggested. (main text: page 6, line 155)

Comment 3-9:

Line 264: “funcitons” to “functions”.

Response 3-9:

Changed as suggested.

Comment 3-10:

Lines 642-649: the following change would improve the english of this sentence, change “..we performed a comparative transcriptomic analysis of seven homologous tissues from pig and nine representative vertebrate mammalian models including... to “we performed comparative analyses of the transcriptomes of seven homologous tissues from pigs and nine representative vertebrate mammalian models including... “

Response 3-10:

Changed as suggested. (main text: page 19, lines 572-573)

Comment 3-11:

Line 678: change “(the most closely rodent models,...)” to “(the most closely related rodent models,...)”

Response 3-11:

Changed as suggested. (main text: page 20, line 604)

Comment 3-12:

Lines 681-682: change “than it was to other non-rodent...” to “than to other non-rodent...”

Response 3-12:

Changed as suggested. (main text: page 20, line 607)

Comment 3-13:

Lines 736-741: it would be appropriate to cite a reference for this hypothesis.

Response 3-13:

As suggested, we have added a reference for the hypothesis.

Comment 3-14:

Line 758: change “economically important products of pig.” to “economically important products.”

Response 3-14:

Changed as suggested. (main text: page 22, line 675)

Comment 3-15:

Line 760: change “pig” to “pigs”.

Response 3-15:

Changed as suggested. (main text: page 22, lines 676)

Comment 3-16:

Line 790: it may be worth adding a topical note that human ACE2 has been shown to act as a receptor for SARS-CoV-2.

Response 3-16:

Thanks for this constructive suggestion. We have added a brief description into the sentence, as follows:

“ACE2, the SARS-CoV-2 receptor required for cell entry in humans and a target of diabetes therapy, ...” (main text: page 23, lines 707-708)

Comment 3-17:

Line 795: change “pig” to “pigs”

Response 3-17:

Changed as suggested. (main text: page 23, line 713)

Comment 3-18:

Line 926: change “Ensemble” to “Ensembl”.

Response 3-18:

Changed as suggested. (main text: page 29, lines 854)

Comment 3-19:

Figure 1c suggests that the correlation in expression patterns between a) PK15 and kidney and b) PIEC and small intestine or colon are 0.8 or less and that these two cell lines have expression patterns more similar to ovary and uterus than the tissues from which these cell lines were developed. These comparisons are worth some comment in the manuscript.

Response 3-19:

Thanks for this suggestion. We have provided descriptions comparing the expression pattern between these two cell lines and other tissues.

“Notably, PK15 and PIEC cell lines were more similar to epithelial- and endothelium-rich internal tissues (typically, ovary and uterus) than to nervous tissue, muscles, and adipose tissues.” (main text: page 40, lines 1199-1201)

Comment 3-20:

A footnote to Table 1 should be added to confirm which annotation of Sscrofa11.1 is being cited. Based on the number of lncRNA cited (i.e. 6,797) it appears to be Ensembl release 100, April 2020, Ensembl release 101, August 2020 or Ensembl release 102 as stated in the rebuttal letter. However, in all of these releases the number of PCGs is 21,303, not 21,280.

Response 3-20:

Thank you for pointing out this inadvertent omission. We used the annotation release 102 of the Sscrofa11.1. As suggested, we have added a footnote to **Table 1** and revised the number of PCGs from 21,280 to 21,303.